# ASK, AND IT SHALL BE GIVEN: ON THE TURING COMPLETENESS OF PROMPTING

**Ruizhong Qiu, Zhe Xu, Wenxuan Bao, & Hanghang Tong**
University of Illinois Urbana–Champaign
`{rq5,zhexu3,wbao4,htong}@illinois.edu`

## ABSTRACT

Since the success of GPT, large language models (LLMs) have been revolutionizing machine learning and have initiated the so-called *LLM prompting* paradigm. In the era of LLMs, people train a single general-purpose LLM and provide the LLM with different *prompts* to perform different tasks. However, such empirical success largely lacks theoretical understanding. Here, we present the first theoretical study on the LLM prompting paradigm to the best of our knowledge. In this work, we show that prompting is in fact Turing-complete: there exists a finite-size Transformer such that for any computable function, there exists a corresponding prompt following which the Transformer computes the function. Furthermore, we show that even though we use only a single finite-size Transformer, it can still achieve nearly the same complexity bounds as that of the class of all unbounded-size Transformers. Overall, our result reveals that prompting can enable a single finite-size Transformer to be efficiently universal, which establishes a theoretical underpinning for prompt engineering in practice.

## 1 INTRODUCTION

The mainstream architecture of large language models (LLMs; e.g., OpenAI, 2024; Anthropic, 2024; Meta, 2024; Google, 2024) is Transformers (Vaswani et al., 2017). There has been a series of theoretical studies on Transformers under realistic abstractions (Pérez et al., 2019; Bhattamishra et al., 2020; Hahn, 2020; Pérez et al., 2021; Hao et al., 2022; Liu et al., 2023a; Chiang et al., 2023; Merrill & Sabharwal, 2023; Roberts, 2023; Merrill & Sabharwal, 2024a;b; Hou et al., 2024; Li et al., 2024). For example, Pérez et al. (2021) have shown that the class of all Transformers with $\mathrm{hardmax}$ attention is Turing-complete: for any computable function $\varphi \in \mathsf{TIME}_1(t(n))$, there exists a Transformer that computes $\varphi$ using $\mathrm{O}(t(n))$ chain-of-thought (CoT; Wei et al., 2022b) steps and $\mathrm{O}(\log(n + t(n)))$ precision on length-$n$ inputs; Merrill & Sabharwal (2024a) have later improved the CoT complexity to $\mathrm{O}(t(n))$ for $\mathsf{TIME}(t(n))$ functions. These works have finely characterized the capacities and limits of Transformers under the classic *one-model-one-task* paradigm.

Nevertheless, existing theoretical studies fail to align with the *LLM prompting* practice (i.e., *one-model-many-tasks*). In the era of LLMs, people train a single general-purpose LLM and provide the LLM with different *prompts* to perform different tasks. Since the success of GPT (Brown et al., 2020), the LLM prompting paradigm has revolutionized machine learning (Liu et al., 2023b). For example, a bonus capability arising from prompting is zero-shot learning (Wei et al., 2022a): when provided with suitable prompts, LLMs can even perform novel tasks not present in their training corpora. Such empirical success calls for a theoretical understanding of the LLM prompting paradigm:

*Fundamentally, how powerful is the LLM prompting paradigm?*

We answer this call and present the first theory on the LLM prompting paradigm to the best of our knowledge. In this work, we show that prompting is in fact *Turing-complete*: there exists a finite-size Transformer such that for any computable function, there exists a corresponding prompt following which the Transformer computes the function. Furthermore, we show that prompting is not only universal but also *efficiently* universal: even though we use one finite-size Transformer, it can still achieve nearly the same complexity bounds as that of the class of all unbounded-size Transformers.

**Main contributions.** Our main contributions are informally stated as follows:

- **Expressive power.** We show that prompting is Turing-complete: there exists a finite-size Transformer $\Gamma$ such that for any computable function $\varphi$, there exists a finite prompt $\boldsymbol{\pi}_\varphi$ such that for any input $\boldsymbol{x}$, the Transformer $\Gamma$ computes $\varphi(\boldsymbol{x})$ following the prompt $\boldsymbol{\pi}_\varphi$. Importantly, our constructed Transformer $\Gamma$ is independent of the function $\varphi$, the prompt $\boldsymbol{\pi}_\varphi$ is independent of the input $\boldsymbol{x}$, and the input $\boldsymbol{x}$ can be arbitrarily long.

- **Simple construction.** In fact, it is not hard to deduce the existence of such a Transformer $\Gamma$ by combining Theorem 1 of Hennie & Stearns (1966) and Theorem 3.4 of Pérez et al. (2019), but explicitly constructing the Transformer via that approach is cumbersome. Instead, we provide a simple constructive proof, which makes it easy to further study the properties of the construction such as CoT complexity and precision complexity.

- **CoT complexity.** We show that our $\Gamma$ can compute any $\mathsf{TIME}_2(t(n))$ function within $\mathrm{O}(t(n))$ CoT steps and can compute any $\mathsf{TIME}(t(n))$ function within $\mathrm{O}(t(n)\log t(n))$ CoT steps for any length-$n$ input. Notably, our result shows that even a *single* Transformer can still achieve nearly the same CoT complexity as the class of *all* Transformers does.

- **Precision complexity.** We show that our $\Gamma$ can compute any $\mathsf{TIME}(t(n))$ function within $\mathrm{O}(\log(n + t(n)))$ bits of precision for any length-$n$ input. Notably, our result shows that even a *single* Transformer can still achieve the same precision complexity as the class of *all* Transformers does. In particular, $\Gamma$ can decide any P language within *log-precision*.

## 1.1 RELATED WORK

In modern machine learning (Wei et al., 2024; Chen et al., 2024; Liu et al., 2024a;b;c; 2023c; Qiu et al., 2024b;a; 2023; 2022; Xu et al., 2024; Qiu & Tong, 2024; Zeng et al., 2024; Lin et al., 2024; Yoo et al., 2025; 2024; Chan et al., 2024; Wu et al., 2024; He et al., 2024; Wang et al., 2023), Transformers have nowadays become a mainstream architecture. Existing theoretical studies on Transformers fall under the classic *one-model-one-task* paradigm: they need to construct different Transformers for different tasks. There are two lines of related work: (i) when at most $\mathrm{O}(1)$ CoT steps are allowed, it has been shown that Transformers are capable but far from Turing-complete (Hahn, 2020; Hao et al., 2022; Liu et al., 2023a; Chiang et al., 2023; Merrill & Sabharwal, 2023; 2024b); (ii) when more CoT steps are allowed, it has been shown that the expressive power of Transformers increases with the number of CoT steps (Pérez et al., 2019; Bhattamishra et al., 2020; Pérez et al., 2021; Roberts, 2023; Merrill & Sabharwal, 2024a; Hou et al., 2024; Li et al., 2024). Besides that, there have recently been studies on the learnability (Malach, 2023; Grau-Moya et al., 2024) and the in-context learning capability (Akyürek et al., 2022; von Oswald et al., 2023; Zhang et al., 2024; Ahn et al., 2024; Vladymyrov et al., 2024). Nevertheless, no existing work studies the LLM prompting paradigm (i.e., the one-model-many-tasks paradigm). Our work is the first to bridge this gap to the best of our knowledge.

## 1.2 TECHNICAL OVERVIEW

A core step of our constructive proof is to construct a new *model of computation* (called 2-PTMs) that can be easily encoded into a prompt using a finite alphabet. Furthermore, we show that 2-PTMs are not only Turing-complete but also nearly as efficient as Turing machines.

**Theorem** (informal version of Theorem 4.1). *Any* $\mathsf{TIME}(t(n))$ *function can be computed by a 2-PTM within* $\mathrm{O}(t(n)\log t(n))$ *steps.* ☐

Given any computable function $\varphi$, we encode its 2-PTM into a prompt $\boldsymbol{\pi}_\varphi$. Then, it remains to construct a Transformer $\Gamma$ that can execute 2-PTMs. Since it is known that Transformers without CoTs are not universal (Hahn, 2020), the Transformer $\Gamma$ needs to use CoT steps to execute 2-PTMs. Specifically, we use CoT steps to record the execution steps of the 2-PTM so that the Transformer can restore the state of the 2-PTM at any step. This establishes the CoT complexity of $\Gamma$.

**Corollary** (informal version of Corollary 4.5). *Our constructed* $\Gamma$ *can compute any* $\mathsf{TIME}(t(n))$ *function within* $\mathrm{O}(t(n)\log t(n))$ *CoT steps.* ☐

To incorporate input $\boldsymbol{x}$ into computation, we use $\mathrm{O}(|\boldsymbol{x}|)$ CoT steps to emulate an *imaginary* process of writing the input $\boldsymbol{x}$ onto a tape of the 2-PTM. This implies the precision complexity of $\Gamma$.

**Corollary** (informal version of Corollary 4.7). *Our constructed* $\Gamma$ *can compute any* $\mathsf{TIME}(t(n))$ *function within* $\mathrm{O}(\log(n + t(n)))$ *bits of precision.* ☐

Finally, we construct a decoder-only Transformer that achieves the desiderata above by leveraging *ReLU activation*, *layer normalization*, and *causal attention*.

## 2 PRELIMINARIES

Let $\epsilon$ denote the empty string. Given an alphabet $\Sigma$, let $\Sigma^n$ ($n \geq 0$) denote the set of length-$n$ strings over $\Sigma$, let $\Sigma^* := \bigcup_{n \geq 0} \Sigma^n$ denote the set of all finite strings over $\Sigma$, let $\Sigma^+ := \bigcup_{n \geq 1} \Sigma^n$ denote the set of all non-empty finite strings over $\Sigma$, and let $\Sigma^\omega$ denote the set of all countably infinite strings over $\Sigma$. Each symbol in $\Sigma$ is called a *token*. For a string $\boldsymbol{x} \in \Sigma^*$, let $|\boldsymbol{x}|$ denote the length of the string, and let $\boldsymbol{x}^n$ denote repeating the string $n$ times. For two strings $\boldsymbol{x}, \boldsymbol{y} \in \Sigma^*$, let $\boldsymbol{x} \cdot \boldsymbol{y}$ denote their concatenation. Given a string $\boldsymbol{x} = x_0 \cdot x_1 \cdots x_{|\boldsymbol{x}|-1} \in \Sigma^*$, for an index $i \in \mathbb{Z}$, let $x_i$ denote the ($i \bmod |\boldsymbol{x}|$)-th token of $\boldsymbol{x}$; for indices $i, j \in \mathbb{Z}$ with $i \bmod |\boldsymbol{x}| \leq j \bmod |\boldsymbol{x}|$, let $\boldsymbol{x}_{i:j}$ denote the substring $x_{i \bmod |\boldsymbol{x}|} \cdot x_{(i \bmod |\boldsymbol{x}|)+1} \cdots x_{(j \bmod |\boldsymbol{x}|)-1}$.

### 2.1 THEORY OF COMPUTATION

**Turing machines.** *Turing machines* (TMs; Turing, 1937a) are an abstract model of computation that defines the notion of computability (Church, 1936; Kleene, 1936; Turing, 1937b). An $m$-tape TM is defined by a septuple $(Q, q_{\text{start}}, q_{\text{halt}}, \Sigma, \llcorner, m, \delta)$ where $Q$ is the finite set of states, $q_{\text{start}} \in Q$ is the initial state, $q_{\text{halt}}$ is the halting state, $\Sigma$ is the finite alphabet of the tapes, $\llcorner \in \Sigma$ is the blank symbol, $m \in \mathbb{N}$ is the number of tapes, and $\delta : (Q \setminus \{q_{\text{halt}}\}) \times \Sigma^m \rightharpoonup Q \times (\Sigma \times \{\texttt{L}, \texttt{S}, \texttt{R}\})^m$ is the transition function (L: left; S: stay; R: right). See, e.g., Arora & Barak (2009) for details. By the Church–Turing thesis (Church, 1936; Kleene, 1936; Turing, 1937b), a function $\varphi : \text{dom}\,\varphi \to \{\texttt{0}, \texttt{1}\}^*$ is said to be *computable* if there exists a TM that computes $\varphi(\boldsymbol{x})$ for all inputs $\boldsymbol{x} \in \text{dom}\,\varphi$.

Shannon (1956) has shown that any TM $M$ over any alphabet can be simulated by a TM $M'$ over the binary alphabet $\Sigma = \{\texttt{0}, \texttt{1}\}$ with $\llcorner = \texttt{0}$ (although $M'$ uses more states). Hence, we will assume that TMs have a binary alphabet for simplicity throughout this paper.

**Time complexity.** Let $n$ denote the length of the input. A function $t(n) : \mathbb{N} \to \mathbb{R}_+$ is said to be a *complexity function* iff $t(n)$ is non-decreasing and either $t(n)$ is a constant $> 1$ or $t(n) \to \infty$ as $n \to \infty$. In this work, we refer to the time complexity as the the time complexity on a random-access machine (RAM) unless otherwise stated (e.g., when working with TMs). If $\Theta(t(n))$ is a complexity function, let $\mathsf{TIME}_m(t(n))$ denote the class of functions that can be computed by an $m$-tape TM within $\mathrm{O}(t(n))$ steps; let $\mathsf{TIME}(t(n)) := \bigcup_{m \geq 1} \mathsf{TIME}_m(t(n))$ denote the class of functions that can be computed by a TM within $\mathrm{O}(t(n))$ steps; and let $\mathsf{P} \subset \mathsf{TIME}(\text{poly}(n))$ denote the class of (indicator functions of) languages that can be decided in polynomial time.

### 2.2 NEURAL NETWORKS

**ReLU neural networks.** A ReLU neural network $\psi : \mathbb{R}^{e_0} \to \mathbb{R}^{e_L}$ is a composition $\psi := \psi_{L-1} \circ \cdots \circ \psi_0$ of basic operations $\psi_l : \mathbb{R}^{e_l} \to \mathbb{R}^{e_{l+1}}$ ($l = 0, \ldots, L-1$) where each basic operation $\psi_l(\boldsymbol{z})$ ($\boldsymbol{z} \in \mathbb{R}^{e_l}$) is either an affine map $\psi_l(\boldsymbol{z}) := \boldsymbol{W}_l \boldsymbol{z} + \boldsymbol{b}_l$ ($\boldsymbol{W}_l \in \mathbb{R}^{e_{l+1} \times e_l}$, $\boldsymbol{b}_l \in \mathbb{R}^{e_{l+1}}$), an entry-wise ReLU activation $\psi_l(\boldsymbol{z}) := \max\{\boldsymbol{z}, \boldsymbol{0}\}$ (Fukushima, 1969) with $e_{l+1} = e_l$, or a layer normalization $\psi_l(\boldsymbol{z}) := \frac{\boldsymbol{z}}{\|\boldsymbol{z}\|_2} \mathbb{1}_{[\boldsymbol{z} \neq \boldsymbol{0}]}$ (Ba et al., 2016) with $e_{l+1} = e_l$.

**Decoder-only Transformers.** Mainstream LLMs are based on *decoder-only* Transformers (Radford et al., 2018). Given a finite token alphabet $\Sigma$, a decoder-only Transformer $\Gamma : \Sigma^+ \to \Sigma$ (with greedy decoding) is a composition $\Gamma := \arg\max \circ \Gamma_{\text{out}} \circ \Gamma_{L-1} \circ \cdots \circ \Gamma_0 \circ \Gamma_{\text{emb}}$ consists of an *embedding* layer $\Gamma_{\text{emb}} : \Sigma^+ \to (\mathbb{R}^d)^+$, *causal-attention* Transformer layer $\Gamma_l : (\mathbb{R}^d)^+ \to (\mathbb{R}^d)^+$ ($l = 0, \ldots, L-1$), and an *output* layer $\Gamma_{\text{out}} : \mathbb{R}^d \to \mathbb{R}^\Sigma$. A *token embedding* is a map $\text{emb} : \Sigma \to \mathbb{R}^d$, and a *positional encoding* is a map $\text{pos} : \mathbb{N} \to \mathbb{R}^d$. Following the common practice (Vaswani et al., 2017), given the input token sequence $\boldsymbol{v} = v_0 \cdots v_{|\boldsymbol{v}|-1} \in \Sigma^+$, the embedding layer $\Gamma_{\text{emb}}$ adds a positional encoding $\text{pos}(i)$ to each token embedding $\text{emb}(v_i)$:

$$\boldsymbol{z}_{0,i} := \text{emb}(v_i) + \text{pos}(i), \qquad i = 0, \ldots, |\boldsymbol{v}| - 1, \tag{1}$$

where $\text{pos}$ is a computable function. Following existing theoretical works (e.g., Pérez et al., 2019; Hao et al., 2022; Merrill & Sabharwal, 2024a), we use $\mathrm{hardmax}$ attention as a realistic abstraction of $\mathrm{softmax}$. Given an $\mathbb{R}$ sequence $\boldsymbol{s}$, if the maximum value of $\boldsymbol{s}$ appears $t$ times, then one has

$\text{hardmax}_i(\boldsymbol{s}) := \frac{1}{t}$ if $s_i$ equals the maximum value and $\text{hardmax}_i(\boldsymbol{s}) := 0$ otherwise. Each Transformer layer $\Gamma_l$ has $H_l$ *attention heads* followed by a ReLU neural network $\phi_l : \mathbb{R}^d \to \mathbb{R}^d$. Each attention head $k$ has a *query* map $\text{qry}_{l,k} : \mathbb{R}^d \to \mathbb{R}^{d_{l,k}}$, a *key* map $\text{key}_{l,k} : \mathbb{R}^d \to \mathbb{R}^{d_{l,k}}$, a *value* map $\text{val}_{l,k} : \mathbb{R}^d \to \mathbb{R}^d$, and a *similarity* map $\text{sim}_{l,k} : \mathbb{R} \to \mathbb{R}$ ($k = 0, \ldots, H_l - 1$), all of which are ReLU neural networks. The similarity scores between two tokens $v_i$ and $v_j$ are

$$s_{l,k,i,j} := \text{sim}_{l,k}(\text{qry}_{l,k}(\boldsymbol{z}_{l,i})^\mathsf{T} \text{key}_{l,k}(\boldsymbol{z}_{l,j})). \tag{2}$$

A decoder-only Transformer layer $\Gamma_l$ is computed via *causal attention* and *residual connection*:

$$\boldsymbol{z}_{l+1,i} := \boldsymbol{z}_{l,i} + \phi_l\left(\sum_{k=0}^{H_l-1}\left(\sum_{j=0}^{i} \text{hardmax}_j(s_{l,k,i,0}, \ldots, s_{l,k,i,i}) \, \text{val}_{l,k}(\boldsymbol{z}_{l,j})\right)\right). \tag{3}$$

The output token is greedily selected according to the final outputs of the last token $v_{|\boldsymbol{v}|-1}$:

$$\Gamma(\boldsymbol{v}) := \arg\max_{c \in \Sigma} \Gamma_{\text{out}}(\boldsymbol{z}_{L,|\boldsymbol{v}|-1})_c, \tag{4}$$

where the output layer $\Gamma_{\text{out}}$ is a ReLU neural network.

Given a Transformer $\Gamma$, let $\text{generate}_\Gamma : \Sigma^+ \to \Sigma^+ \cup \Sigma^\omega$ denote autoregressive generation using $\Gamma$. Specifically, let $\boldsymbol{w} \in \Sigma^+$ denote the input sequence and $\boldsymbol{y} \in \Sigma^*$ the (current) output sequence. Initially, $\boldsymbol{y}$ is an empty string. In each generation step, we append $\Gamma(\boldsymbol{w} \cdot \boldsymbol{y})$ to $\boldsymbol{y}$. The generation process ends once the last token of $\boldsymbol{y}$ is a so-called *stop token* $\$ \in \Sigma$, and we define $\text{generate}_\Gamma(\boldsymbol{w}) := \boldsymbol{y}$ at last. It is possible that generation never ends, in which case we have $\boldsymbol{y} \in \Sigma^\omega$. The autoregressive generation procedure is formally presented in Algorithm 1. A Transformer is said to use *log-precision* for a computable function $\varphi$ if the intermediate computation of all generation steps uses $O(\log n)$ bits of precision for every length-$n$ input of function $\varphi$ (Merrill & Sabharwal, 2023).

## 3 TURING COMPLETENESS OF PROMPTING

**Theorem 3.1** (Turing completeness of prompting). *There exist a finite alphabet $\Sigma$, a finite-size decoder-only Transformer $\Gamma : \Sigma^+ \to \Sigma$, and coding schemes* $\text{tokenize} : \{0,1\}^* \to \Sigma^*$ *and* $\text{readout} : \Sigma^* \to \{0,1\}^*$ *with which prompting is Turing-complete: for every computable function* $\varphi : \text{dom}\,\varphi \to \{0,1\}^*$ *with* $\text{dom}\,\varphi \subseteq \{0,1\}^*$, *there exists a prompt* $\boldsymbol{\pi}_\varphi \in \Sigma^+$ *such that for every input* $\boldsymbol{x} \in \text{dom}\,\varphi$, $\text{generate}_\Gamma(\boldsymbol{\pi}_\varphi \cdot \text{tokenize}(\boldsymbol{x}))$ *computes[1] a finite CoT, and*

$$\text{readout}(\text{generate}_\Gamma(\boldsymbol{\pi}_\varphi \cdot \text{tokenize}(\boldsymbol{x}))) = \varphi(\boldsymbol{x}). \tag{5}$$

*Here,* $\Sigma, \Gamma, \text{tokenize}, \text{readout}$ *are independent of* $\varphi$; $\boldsymbol{\pi}_\varphi$ *is independent of* $\boldsymbol{x}$; *for any input* $\boldsymbol{x} \in \{0,1\}^*$, $\text{tokenize}$ *and* $\text{readout}$ *run in* $O(|\boldsymbol{x}|)$ *and* $O(|\varphi(\boldsymbol{x})|)$ *time on a RAM, respectively.*

Our Theorem 3.1 shows that prompting can enable a single Transformer to be universal and establishes a theoretical underpinning for prompt engineering in practice. Note that CoT is necessary for Turing completeness because it is known that Transformers without CoTs cannot even compute the parity function (Hahn, 2020). As a CoT typically contains more information than the answer $\varphi(\boldsymbol{x})$ alone, we need a map $\text{readout} : \Sigma^* \to \{0,1\}^*$ here to extract the answer, resembling the fact that humans need to read out the answer from the generated CoT.

Furthermore, to elucidate the technical non-triviality of our Theorem 3.1, we remark that our Theorem 3.1 has ruled out trivial possibilities:

- **Memorization?** It is impossible that the Transformer $\Gamma$ simply memorizes all computable functions, because there are *infinitely* many computable functions while $\Gamma$ has only a *finite* size (i.e., it has a *finite* number of *finite*-bit parameters).

- **Self-answering?** It is impossible that the prompt $\boldsymbol{\pi}_\varphi$ simply reveals the answers for all possible inputs, because there can be *infinitely* many inputs while the prompt is *finite*.

- **Tautology?** It is impossible that $\Gamma$ simply restates $\boldsymbol{\pi}_\varphi \cdot \text{tokenize}(\boldsymbol{x})$ and lets $\text{tokenize}$ or $\text{readout}$ compute $\varphi(\boldsymbol{x})$ instead, because the time hierarchy theorem (Hartmanis & Stearns, 1965) implies that a computable function $\varphi$ in general can require more than $\Omega(\max\{|\boldsymbol{x}|, |\varphi(\boldsymbol{x})|\})$ time while $\text{tokenize}$ and $\text{readout}$ run in only $O(|\boldsymbol{x}|)$ time and $O(|\varphi(\boldsymbol{x})|)$ time, respectively.

---

[1]Following prior work (e.g., Pérez et al., 2019), we assume that every arithmetic computation in $\Gamma$ is exact.

Our proof of Theorem 3.1 is constructive. In the rest of this section, we will present the core constructions in our proof, including the prompt, the CoT steps, the input tokenizer, and the Transformer. Due to space limit, the complete proof is deferred to Appendix B.

## 3.1 CONSTRUCTION OF PROMPTS

In this subsection, we show how to construct a prompt $\boldsymbol{\pi}_\varphi$ for any given computable function $\varphi$. The basic idea here is that we let the prompt encode a formal description of the function $\varphi$ and later construct a Transformer that can execute the description. The construction of the Transformer is deferred to Section 3.4.

As computability means that there exists a TM that computes $\varphi$, one might seek to encode the TM into the prompt. Unfortunately, a TM can have an unbounded number of states and an unbounded number of tapes, but we are only allowed to construct all prompts using a *single* finite alphabet $\Sigma$ and to execute all prompts using a *single* finite-size Transformer. Even though one can use sophisticated schemes to encode TMs (Turing, 1937a), it remains highly non-trivial to construct a Transformer to efficiently execute such encoded TMs.

Hence, instead of working with TMs directly, here we want a model of computation that (i) can be easily encoded by a single finite alphabet, that (ii) is still Turing-complete, and that (iii) is nearly as efficient as TMs. Although a possible approach is to use an imperative model of computation such as Wang machines (Wang, 1957) and Davis–Sigal–Weyuker's Post–Turing machines (DSW-PTMs; Davis et al., 1994), it is still unknown how to efficiently simulate arbitrary TMs using Wang machines or Davis–Sigal–Weyuker's Post–Turing machines. This suggests that these models of computation might not be the best candidate for constructing the prompt.

To fulfill our desiderata, we propose a new imperative model of computation that extends Wang machines and DSW-PTMs. Inspired by the Hennie–Stearns theorem (Hennie & Stearns, 1966), we let our model of computation use two bi-infinite tapes A and B. Each tape has infinitely many cells over the binary alphabet $\{0, 1\}$ and a head pointing to a cell. We call this model *two-tape Post–Turing machines* (2-PTMs). A 2-PTM is defined by a finite instruction sequence $\boldsymbol{\iota} = \langle \iota_0, \iota_1, \ldots, \iota_{|\boldsymbol{\iota}|-1} \rangle$ where each instruction $\iota_j$ ($0 \leq j < |\boldsymbol{\iota}|$) is one of the following:

- #: halt;
- $\tau$L ($\tau \in \{A, B\}$): move the head for tape $\tau$ one cell left, and go to $\iota_{j+1}$;
- $\tau$R ($\tau \in \{A, B\}$): move the head for tape $\tau$ one cell right, and go to $\iota_{j+1}$;
- $\tau$0 ($\tau \in \{A, B\}$): write 0 to the pointed cell of tape $\tau$, and go to $\iota_{j+1}$;
- $\tau$1 ($\tau \in \{A, B\}$): write 1 to the pointed cell of tape $\tau$, and go to $\iota_{j+1}$;
- $\tau!_k$ ($\tau \in \{A, B\}$; $k \neq j$): if the pointed cell of tape $\tau$ is 0, go to $\iota_k$; else, go to $\iota_{j+1}$;
- $\tau?_k$ ($\tau \in \{A, B\}$; $k \neq j$): if the pointed cell of tape $\tau$ is 1, go to $\iota_k$; else, go to $\iota_{j+1}$.

Let $\mathcal{I}$ denote the set of all possible instructions. Before execution, the input is written to tape A, and the head for tape A is pointing to the leftmost cell of the input (called cell 0). All blank tape cells are filled with 0. Then, execution starts from instruction $\iota_0$ and halts upon instruction #, and the output is the content left on tape A starting from cell 0. We will show in Section 4.1 that 2-PTMs are not only Turing-complete but also nearly as efficient as TMs.

Next, we describe how to construct a prompt for a given computable function. Recall that 2-PTMs use the binary alphabet $\{0, 1\}$ with blank symbol being 0. To distinguish the input symbol 0 and the blank symbol 0, we employ Shannon's encoding $S : \{0, 1\}^* \to \{0, 1\}^*$ (Shannon, 1956) to translate inputs and outputs of computable functions:

$$S(\epsilon) := \epsilon, \quad S(0) := 10, \quad S(1) := 11; \tag{6}$$

$$S(\boldsymbol{x}) := S(x_0) \cdots S(x_{|\boldsymbol{x}|-1}), \quad \boldsymbol{x} \in \{0, 1\}^*. \tag{7}$$

Thus, we identify 00 as the blank symbol. Note that Shannon's encoding $S$ is injective, and that both $S$ and its corresponding decoding $S^{-1}$ are computable in linear time. Since the class of 2-PTMs is Turing-complete, then given any computable function $\varphi$, there exists a 2-PTM $\boldsymbol{\iota} \in \mathcal{I}^+$ that computes $S(\varphi(\boldsymbol{x}))$ from $S(\boldsymbol{x})$ for all $\boldsymbol{x} \in \mathrm{dom}\,\varphi$. It remains to encode $\boldsymbol{\iota}$ into a prompt $\boldsymbol{\pi}_\varphi$.

We will define a map $P : \mathbb{N} \times \mathcal{I} \to \Sigma^*$ to encode each instruction $\iota_j$ and let the prompt be the concatenation of $P(j, \iota_j)$, where the alphabet $\Sigma$ will be specified later. For instructions #, $\tau$L, $\tau$R,

$\tau$0, $\tau$1, we can simply create a corresponding token in $\Sigma$ for each of them:

$$P(j, \iota_j) := \iota_j \qquad \text{if } \iota_j \text{ is one of } \#, \tau\text{L}, \tau\text{R}, \tau\text{0}, \tau\text{1}. \tag{8}$$

For instructions $\tau!_k$ and $\tau?_k$, since $k$ can be any natural number, we can no longer create a token for each $k$ because otherwise the alphabet $\Sigma$ would be infinite. Instead, to help the Transformer to execute the prompt, we use a unary encoding w.r.t $k - j$:

$$P(j, \iota_j) := \begin{cases} \sigma \cdot -^{j-k} \cdot @ & \text{if } \iota_j = \sigma_k \text{ and } k < j, \\ \sigma \cdot +^{k-j} \cdot @ & \text{if } \iota_j = \sigma_k \text{ and } k > j, \end{cases} \qquad \sigma \in \{\tau!, \tau?\}_{\tau \in \{\text{A,B}\}}, \tag{9}$$

where we create seven auxiliary tokens $\text{A!}, \text{B!}, \text{A?}, \text{B?}, -, +, @ \in \Sigma$ to be used by the Transformer. Finally, we construct the prompt as

$$\boldsymbol{\pi}_\varphi := \hat{} \cdot P(0, \iota_0) \cdots P(|\boldsymbol{\iota}| - 1, \iota_{|\boldsymbol{\iota}|-1}) \cdot \$, \tag{10}$$

where we create two auxiliary tokens $\hat{}, \$ \in \Sigma$ to be used by the Transformer.

**Example.** A 2-PTM for deciding the DYCK language (Schützenberger, 1963) is[2]

`A?₁₄AOALAOALA?₁ARARA1ARBLB?₁₃A1#ARA?₁₉B1BRB!₂₁BLB!₂₄BOARB!₀ALARARA?₂₅AOALAOALA?₂₈ARARA1#,`

and its corresponding prompt is

```
^A?+++++++++++++@AOALAOALA?----@ARARA1ARBLB?++@A1#ARA?++++@B1BRB!+++@BLB!++++@BOARB!----------------------@
ALARARA?--@AOALAOALA?----@ARARA1#$.
```

## 3.2 RECORDING EXECUTION IN CoT STEPS

It is known that finite-size Transformers without CoT steps are not universal (Hahn, 2020). To achieve Turing completeness, here we leverage CoT steps to record execution steps of the 2-PTM $\boldsymbol{\iota}$ so that the Transformer can restore the state of $\boldsymbol{\iota}$ at any execution step. In this subsection, we focus on the case where the input is empty; we will describe how to incorporate the input in Section 3.3.

Let $\iota_j$ denote the current instruction, and let $c_\text{A}, c_\text{B}$ denote the pointed cell of tapes A and B, respectively. Note that each execution step can be summarized as a quadruple $(j, \iota_j, c_\text{A}, c_\text{B})$, where $\iota_j$ is the current instruction, and $c_\text{A}$ and $c_\text{B}$ are the currently pointed cell of tapes A and B, respectively. We will define a map $C : \mathbb{N} \times \mathcal{I} \times \{0, 1\}^2 \to \Sigma^*$ that maps each execution step to one or more CoT steps. If $\iota_j$ is one of $\#, \tau\text{L}, \tau\text{R}, \tau\text{0}, \tau\text{1}$, we simply use a single CoT step to record $\iota_j$:

$$C(j, \iota_j, c_\text{A}, c_\text{B}) := \iota_j \qquad \text{if } \iota_j \text{ is one of } \#, \tau\text{L}, \tau\text{R}, \tau\text{0}, \tau\text{1}. \tag{11}$$

If $\iota_j$ is $\tau!_k$ or $\tau?_k$, we record whether the go-to condition is satisfied or not:

$$C(j, \iota_j, c_\text{A}, c_\text{B}) := \begin{cases} / & \text{if } \iota_j = \tau!_k \text{ and } c_\tau \neq 0; \\ = \cdot -^{j-k} \cdot @ & \text{if } \iota_j = \tau!_k, \ c_\tau = 0, \text{ and } k < j; \\ = \cdot +^{k-j} \cdot @ & \text{if } \iota_j = \tau!_k, \ c_\tau = 0, \text{ and } k > j; \\ / & \text{if } \iota_j = \tau?_k \text{ and } c_\tau \neq 1; \\ = \cdot -^{j-k} \cdot @ & \text{if } \iota_j = \tau?_k, \ c_\tau = 1, \text{ and } k < j; \\ = \cdot +^{k-j} \cdot @ & \text{if } \iota_j = \tau?_k, \ c_\tau = 1, \text{ and } k > j; \end{cases} \tag{12}$$

where we create two auxiliary tokens $/, = \in \Sigma$ to indicate whether the go-to condition is unsatisfied or satisfied, respectively. The execution step stops once it reaches the halting instruction $\#$.

Finally, we append the output $\varphi(\boldsymbol{x})$ after the execution steps in the CoT. We create a new auxiliary token $: \in \Sigma$ to mark the beginning of the output and reuse the token $\$ \in \Sigma$ to mark the end of the output, and we put the output $\varphi(\boldsymbol{x}) \in \{0, 1\}^* \subseteq \Sigma^*$ between $:$ and $\$$. Hence, we define readout as extracting the part of the CoT between $:$ and $\$$ (see Algorithm 2). By construction, the number of CoT steps is proportional to the number of execution steps plus the length of the output.

**Example (cont'd).** An empty input $\epsilon$ is in the DYCK language. Its corresponding CoT steps are:

`/AOALAOAL/ARARA1ARBL/A1:1$.`

To recap, we have created a finite alphabet of 23 tokens all together:

$$\Sigma := \{\#, \text{AL}, \text{BL}, \text{AR}, \text{BR}, \text{A0}, \text{B0}, \text{A1}, \text{B1}, \text{A!}, \text{B!}, \text{A?}, \text{B?}, -, +, @, \hat{}, \$, /, =, :, \text{0}, \text{1}\}. \tag{13}$$

---

[2]From now on, we will omit delimiters for conciseness when there is no ambiguity.

### 3.3 Construction of input tokenizer

In this subsection, we describe our input tokenizer $\text{tokenize} : \{0,1\}^* \to \Sigma^*$, which is independent of the function $\varphi$ and can encode any input $\boldsymbol{x} \in \{0,1\}^*$ without changing the prompt $\boldsymbol{\pi}_\varphi$. Although it is possible to introduce additional auxiliary tokens to represent inputs, here we provide a simpler construction that makes use of only the existing tokens in $\Sigma$.

The key idea here is to use CoT steps to emulate an imaginary process of writing the input to tape A. We will define a map $E : \{0,1\}^+ \to \Sigma^+$ that encodes an input $\boldsymbol{x}$ into CoT steps. Since 2-PTMs assume that the head for tape A is initially pointing to the leftmost cell of the input, we write the input from right to left onto tape A. Hence, for each bit $x_i$ of the input $\boldsymbol{x}$, we write its Shannon encoding $S(x_i)$ from right to left onto tape A:

$$E(\texttt{0}) := \texttt{ALALA1}, \qquad E(\texttt{1}) := \texttt{ALA1ALA1}. \tag{14}$$

For the entire input, we concatenate the CoT steps of each bit from right to left:

$$E(\boldsymbol{x}) := E(x_{|\boldsymbol{x}|-1}) \cdots E(x_0), \qquad \boldsymbol{x} \in \{0,1\}^+. \tag{15}$$

Intuitively, $E(\boldsymbol{x})$ represents execution steps of an imaginary program that writes $S(\boldsymbol{x})$ onto tape A.

To ensure that the tape-A head is at cell $0$ after writing the input, we first move the tape-A head $|S(\boldsymbol{x})| = 2|\boldsymbol{x}|$ steps right. Hence, we define the CoT steps for writing $S(\boldsymbol{x})$ as $Z : \{0,1\}^+ \to \Sigma^+$,

$$Z(\boldsymbol{x}) := \texttt{AR}^{2|\boldsymbol{x}|} \cdot E(\boldsymbol{x}), \qquad \boldsymbol{x} \in \{0,1\}^+. \tag{16}$$

Nevertheless, there is a caveat: a 2-PTM should start from $\iota_0$, but these extra CoT steps $Z(\boldsymbol{x})$ will confuse the Transformer into starting from $\iota_{|Z(\boldsymbol{x})|}$. To address this caveat, our input tokenizer $\text{tokenize}$ additionally employs an imaginary go-to step to let the Transformer go back to $\iota_0$. Following Equation (12), we re-use $\texttt{=}, \texttt{-}, \texttt{@}$ to construct the imaginary go-to step:

$$\text{tokenize}(\boldsymbol{x}) := \begin{cases} \epsilon & \text{if } \boldsymbol{x} = \epsilon, \\ Z(\boldsymbol{x}) \cdot \texttt{=} \cdot \texttt{-}^{|Z(\boldsymbol{x})|} \cdot \texttt{@} & \text{if } \boldsymbol{x} \neq \epsilon. \end{cases} \tag{17}$$

**Example.** Since input $\texttt{01}$ has Shannon encoding $S(\texttt{01}) = \texttt{1011}$, it is tokenized as

$$\text{tokenize}(\texttt{01}) = \texttt{ARARARARALA1ALA1ALALA1=-----------@}.$$

### 3.4 Construction of Transformer

In this subsection, we sketch how to construct a decoder-only Transformer $\Gamma$ that executes prompts through CoT steps as described in Section 3.2. Due to the space limit, we only present three core operations to be used by $\Gamma$ here and defer the detailed construction to Appendix B.

**Boolean algebra via ReLU activation.** Let $\text{ReLU}(z) := \max\{z, 0\}$ denote the ReLU function. Boolean algebra is a basic building block of our Transformer for, e.g., checking the go-to condition. A core operation in Boolean algebra is the $\wedge$ operation. Here, we implement $\wedge$ via ReLU activation:

$$u \wedge v = \text{ReLU}(u + v - 1), \qquad u, v \in \{0, 1\}. \tag{18}$$

Together with negation $\neg v = 1 - v$, they can implement all other Boolean operations such as $u \vee v = \neg((\neg u) \wedge (\neg v))$ and $u \oplus v = (u \wedge (\neg v)) + ((\neg u) \wedge v)$.

**Equality check via layer normalization.** Let $\text{LN}(\boldsymbol{z}) := \frac{\boldsymbol{z}}{\|\boldsymbol{z}\|_2} 1_{[\boldsymbol{z} \neq \boldsymbol{0}]}$ denote layer normalization (LN). When checking whether a tape cell has been written or not, we will need an equality check between two real numbers: $\text{NotEqual} : \mathbb{R}^2 \to \{0, 1\}$ where $\text{NotEqual}(u, v) := 1_{[u \neq v]}$ $(u, v \in \mathbb{R})$. Since $\text{NotEqual}$ is not a continuous map, it cannot be implemented using only affine maps or ReLU activation. Instead, we implement $\text{NotEqual}$ via layer normalization as follows:

$$\text{NotEqual}(u, v) := \text{ReLU}(\text{LN}(u - v)) + \text{ReLU}(\text{LN}(v - u)), \qquad u, v \in \mathbb{R}. \tag{19}$$

**Farthest retrieval via causal attention.** A core operation to be used by the Transformer $\Gamma$ can be abstracted as follows: Given a number sequence $\boldsymbol{v} = \langle v_0, \ldots, v_{|\boldsymbol{v}|-1} \rangle$ with $v_i \in \{-1, 0, +1\}$ for all $i$, find the *smallest* $i$ such that $\sum_{j=0}^{i} v_j = \sum_{j=0}^{|\boldsymbol{v}|-1} v_j$. However, since causal attention is an average

rather than a sum, it cannot even compute $\sum_{j=0}^{i} v_j$ for arbitrary $i$. It can compute the average $\frac{1}{i+1} \sum_{j=0}^{i} v_j$ for every $i$, but $\sum_{j=0}^{i} v_j = \sum_{j=0}^{|\boldsymbol{v}|-1} v_j$ does not mean $\frac{1}{i+1} \sum_{j=0}^{i} v_j = \frac{1}{|\boldsymbol{v}|} \sum_{j=0}^{|\boldsymbol{v}|-1} v_j$. To bypass the mismatched coefficients $\frac{1}{i+1}$ and $\frac{1}{|\boldsymbol{v}|}$, we use LN to remove the averaging coefficient $\frac{1}{i+1}$. First, we let every token attend to the initial token ˆ at $i = 0$ to compute $\frac{1}{i+1}$ and compute

$$\boldsymbol{u}_i := \text{LN}\left(\frac{\sum_{j=0}^{i} v_j}{i+1}, \frac{1}{i+1}\right)^{\mathsf{T}} = \left(\frac{\sum_{j=0}^{i} v_j}{\sqrt{(\sum_{j=0}^{i} v_j)^2 + 1}}, \frac{1}{\sqrt{(\sum_{j=0}^{i} v_j)^2 + 1}}\right)^{\mathsf{T}}. \quad (20)$$

If $\sum_{j=0}^{i} v_j = \sum_{j=0}^{|\boldsymbol{v}|-1} v_j$, we have $\boldsymbol{u}_i^{\mathsf{T}} \boldsymbol{u}_{|\boldsymbol{v}|-1} = 1$; if $\sum_{j=0}^{i} v_j \neq \sum_{j=0}^{|\boldsymbol{v}|-1} v_j$, we have $\boldsymbol{u}_i^{\mathsf{T}} \boldsymbol{u}_{|\boldsymbol{v}|-1} < 1$. Thus, we can add $\boldsymbol{u}_{|\boldsymbol{v}|-1}$ to the query map qry and add $\boldsymbol{u}_i$ to the key map key in attention to retrieve an $i$ with $\sum_{j=0}^{i} v_j = \sum_{j=0}^{|\boldsymbol{v}|-1} v_j$.

It remains to retrieve the *smallest* such $i$ via causal attention. Since $v_i \in \{-1, 0, +1\}$, then note that if $\boldsymbol{u}_i^{\mathsf{T}} \boldsymbol{u}_{|\boldsymbol{v}|-1} < 1$, we in fact have

$$\boldsymbol{u}_i^{\mathsf{T}} \boldsymbol{u}_{|\boldsymbol{v}|-1} < \left(\frac{|\boldsymbol{v}|}{\sqrt{|\boldsymbol{v}|^2 + 1}}, \frac{1}{\sqrt{|\boldsymbol{v}|^2 + 1}}\right)\left(\frac{|\boldsymbol{v}| + 1}{\sqrt{(|\boldsymbol{v}| + 1)^2 + 1}}, \frac{1}{\sqrt{(|\boldsymbol{v}| + 1)^2 + 1}}\right)^{\mathsf{T}} \quad (21)$$

$$= \frac{|\boldsymbol{v}|(|\boldsymbol{v}| + 1) + 1}{\sqrt{|\boldsymbol{v}|^2 + 1}\sqrt{(|\boldsymbol{v}| + 1)^2 + 1}} = 1 - \Omega\left(\frac{1}{|\boldsymbol{v}|^4}\right). \quad (22)$$

This motivates us to use the following quantity, which can be computed in positional encoding pos:

$$p_i := 1 - \frac{(i+1)(i+2) + 1}{\sqrt{(i+1)^2 + 1}\sqrt{(i+2)^2 + 1}} = \Omega\left(\frac{1}{(i+1)^4}\right). \quad (23)$$

Note that if $\boldsymbol{u}_i^{\mathsf{T}} \boldsymbol{u}_{|\boldsymbol{v}|-1} = 1$ and $\boldsymbol{u}_j^{\mathsf{T}} \boldsymbol{u}_{|\boldsymbol{v}|-1} < 1$, we also have

$$\boldsymbol{u}_j^{\mathsf{T}} \boldsymbol{u}_{|\boldsymbol{v}|-1} + \frac{p_{|\boldsymbol{v}|-1}}{j+1} \leq \boldsymbol{u}_j^{\mathsf{T}} \boldsymbol{u}_{|\boldsymbol{v}|-1} + p_{|\boldsymbol{v}|-1} < 1 = \boldsymbol{u}_i^{\mathsf{T}} \boldsymbol{u}_{|\boldsymbol{v}|-1} < \boldsymbol{u}_i^{\mathsf{T}} \boldsymbol{u}_{|\boldsymbol{v}|-1} + \frac{p_{|\boldsymbol{v}|-1}}{i+1}. \quad (24)$$

Therefore, we can retrieve the smallest $i$ with $\boldsymbol{u}_i^{\mathsf{T}} \boldsymbol{u}_{|\boldsymbol{v}|-1} = 1$ by using query vector $(\boldsymbol{u}_{|\boldsymbol{v}|-1}, p_{|\boldsymbol{v}|-1})^{\mathsf{T}}$ and key vector $(\boldsymbol{u}_i, \frac{1}{i+1})^{\mathsf{T}}$ in causal attention.

## 4 COMPLEXITY BOUNDS

We (i) show in Section 4.1 that 2-PTMs are Turing-complete and nearly as efficient as TMs and (ii) use this result to characterize the complexities of our constructed $\Gamma$ in Sections 4.2 & 4.3. Throughout this section, let $t(n)$ denote a complexity function. Following the convention in complexity theory, we allow different computable functions to have different constant factors in big O.

### 4.1 EFFICIENT SIMULATION OF TMs BY 2-PTMs

Since 2-PTMs are an essential component of our construction, we need the complexity bounds of 2-PTMs to analyze the complexities of $\Gamma$. While Wang machines and DSW-PTMs suffer from a polynomial slowdown over TMs (Neary et al., 2014), we show that our 2-PTMs in fact has only at most a logarithmic slowdown over TMs. Let $\text{TIME}_{\text{2-PTM}}(t(n))$ denote the class of functions that can be computed by a 2-PTM within $O(t(n))$ steps.

**Theorem 4.1** (efficient multi-tape simulation). $\text{TIME}(t(n)) \subseteq \text{TIME}_{\text{2-PTM}}(t(n) \log t(n))$.

Theorem 4.1 shows that our 2-PTMs are Turing-complete and nearly as efficient as TMs. To prove it, we need the following Lemma 4.2 to establish a relation between two-tape TMs and 2-PTMs.

**Lemma 4.2** (two-tape simulation). $\text{TIME}_2(t(n)) \subseteq \text{TIME}_{\text{2-PTM}}(t(n))$.

*Proof of Lemma 4.2.* For any computable function $\varphi \in \text{TIME}_2(t(n))$, there is a two-tape TM $M = (Q, q_{\text{start}}, q_{\text{halt}}, \{0, 1\}, \text{␣} = 0, m = 2, \delta)$ that maps $S(\boldsymbol{x})$ to $S(\varphi(\boldsymbol{x}))$ within time $O(t(n))$, according to Shannon (1956). Suppose w.l.o.g. that $Q = \{0, 1, \ldots, K\}$ $(K \in \mathbb{N}_+)$ and that $q_{\text{start}} = 0, q_{\text{halt}} = K$. We will construct a 2-PTM $\boldsymbol{\iota}$ of length $|\boldsymbol{\iota}| = 27K + 1$ that simulates $M$ within time $O(t(n))$.

Recall that $\delta : (Q \setminus \{q_{\text{halt}}\}) \times \{0, 1\}^2 \to Q \times (\{0, 1\} \times \{\text{L}, \text{S}, \text{R}\})^2$. We use exactly six instructions $\boldsymbol{\eta}_{(q', c_{\text{A}}, d_{\text{A}}, c_{\text{B}}, d_{\text{B}})}$ to simulate each transition $(q', c'_{\text{A}}, d_{\text{A}}, c'_{\text{B}}, d_{\text{B}})$, where $q' \in Q$, $c'_{\text{A}}, c'_{\text{B}} \in \{0, 1\}$, $d_{\text{A}}, d_{\text{B}} \in \{\text{L}, \text{S}, \text{R}\}$. If $d_{\text{A}}, d_{\text{B}} \neq \text{S}$, we let

$$\boldsymbol{\eta}_{(q', c'_{\text{A}}, d_{\text{A}}, c'_{\text{B}}, d_{\text{B}})} := \langle \text{A}c'_{\text{A}}, \text{A}d_{\text{A}}, \text{B}c'_{\text{B}}, \text{B}d_{\text{B}}, \text{A} !\, _{27q'}, \text{A} ?\, _{27q'} \rangle. \tag{25}$$

If $d_{\text{A}} = \text{S}$ or $d_{\text{B}} = \text{S}$, we can replace the corresponding $\text{A}d_{\text{A}}$ or $\text{B}d_{\text{B}}$ with $\text{A}c'_{\text{A}}$ or $\text{B}c'_{\text{B}}$, respectively, so that $\boldsymbol{\eta}_{(q', c'_{\text{A}}, d_{\text{A}}, c'_{\text{B}}, d_{\text{B}})}$ still has exactly six instructions.

Then for each $q \in Q \setminus \{q_{\text{halt}}\}$, we use 27 instructions to simulate its transition rules:

$$\boldsymbol{\iota}_{27q:27q+27} := \langle \text{A} ?\, _{27q+14}, \text{B} ?\, _{27q+8}, \boldsymbol{\eta}_{\delta(q,0,0)}, \boldsymbol{\eta}_{\delta(q,0,1)}, \text{B} ?\, _{27q+21}, \boldsymbol{\eta}_{\delta(q,1,0)}, \boldsymbol{\eta}_{\delta(q,1,1)} \rangle. \tag{26}$$

For the halting state $q_{\text{halt}} = K$, we use one instruction $\boldsymbol{\iota}_{27K} := \#$ to simulate it.

Since $\boldsymbol{\iota}$ simulates each transition of $M$ by $\text{O}(1)$ steps, then $\boldsymbol{\iota}$ also has time complexity $\text{O}(t(n))$.

Due to space limit, the proof of correctness of $\boldsymbol{\iota}$ is deferred to Appendix A. $\qquad\square$

We are now ready to prove Theorem 4.1.

*Proof sketch of Theorem 4.1.* Let $\varphi \in \text{TIME}(t(n))$. Then, there exists a TM $M$ that computes $\varphi$ within $T(n) = \text{O}(t(n))$ steps for every length-$n$ input.

*Case 1:* There exists $n_0 \in \mathbb{N}$ such that $T(n_0) < n_0$. Thus, the behavior of the TM $M$ depends only on the first $T(n_0) < n_0$ bits of the input. Hence, further increasing the length of the input does not change the behavior of $M$. Therefore, a tighter time complexity $\widetilde{T}(n)$ of $M$ is

$$\widetilde{T}(n) \leq T(n_0) = \text{O}(1), \qquad \forall n \in \mathbb{N}. \tag{27}$$

This implies that $\varphi \in \text{TIME}_2(1) \subseteq \text{TIME}_{\text{2-PTM}}(1) \subseteq \text{TIME}_{\text{2-PTM}}(t(n) \log t(n))$.

*Case 2:* $T(n) \geq n$ for all $n \in \mathbb{N}$. Then by the Hennie–Stearns theorem (Hennie & Stearns, 1966), $\varphi \in \text{TIME}_2(T(n) \log T(n))$. Therefore, by Lemma 4.2,

$$\varphi \in \text{TIME}_2(t(n) \log t(n)) \subseteq \text{TIME}_{\text{2-PTM}}(t(n) \log t(n)). \tag{28}$$

It follows from the above two cases that $\text{TIME}(t(n)) \subseteq \text{TIME}_{\text{2-PTM}}(t(n) \log t(n))$. $\qquad\square$

Theorem 4.1 shows that 2-PTMs have only at most a logarithmic slowdown over TMs. In subsequent Sections 4.2 & 4.3, we will use Theorem 4.1 to characterize the CoT complexity and the precision complexity of our constructed $\Gamma$.

## 4.2 COT COMPLEXITY

In this subsection, we analyze the CoT complexity of our construction. We will show that our constructed $\Gamma$ can compute any $\text{TIME}_2(t(n))$ function within $\text{O}(t(n))$ CoT steps and any $\text{TIME}(t(n))$ function within $\text{O}(t(n) \log t(n))$ CoT steps for any length-$n$ input.

**Definition 4.3** (CoT complexity class). *Let $\text{CoT}_{\Gamma}(t(n))$ be the class of functions that our constructed Transformer $\Gamma$ can compute within $\text{O}(t(n))$ CoT steps.*

**Lemma 4.4** (CoT complexity for 2-PTMs). $\text{TIME}_{\text{2-PTM}}(t(n)) \subseteq \text{CoT}_{\Gamma}(t(n))$.

*Proof sketch.* For any $\varphi \in \text{TIME}_{\text{2-PTM}}(t(n))$, since prompt $\boldsymbol{\pi}_{\varphi}$ has length $|\boldsymbol{\pi}_{\varphi}| = \text{O}(1)$, then by Section 3.2, each $\#, \tau<, \tau>, \tau 0, \tau 1$ takes $1 = \text{O}(1)$ CoT step, and each $\tau !\, _k, \tau ?\, _k$ takes at most $\text{O}(|\boldsymbol{\pi}_{\varphi}|) = \text{O}(1)$ CoT steps. This implies that $\text{TIME}_{\text{2-PTM}}(t(n)) \subseteq \text{CoT}_{\Gamma}(t(n))$. $\qquad\square$

**Corollary 4.5** (CoT complexity for TMs). *For two-tape TMs, $\text{TIME}_2(t(n)) \subseteq \text{CoT}_{\Gamma}(t(n))$. For general TMs, $\text{TIME}(t(n)) \subseteq \text{CoT}_{\Gamma}(t(n) \log t(n))$.*

*Proof.* For two-tape TMs, by Lemmas 4.2 & 4.4,

$$\text{TIME}_2(t(n)) \subseteq \text{TIME}_{\text{2-PTM}}(t(n)) \subseteq \text{CoT}_{\Gamma}(t(n)). \qquad\square$$

For general TMs, by Theorem 4.1 & Lemma 4.4,

$$\text{TIME}(t(n)) \subseteq \text{TIME}_{\text{2-PTM}}(t(n) \log t(n)) \subseteq \text{CoT}_{\Gamma}(t(n) \log t(n)).$$

Corollary 4.5 shows prompting a *single* Transformer can compute any $\mathsf{TIME}_2(t(n))$ function within $\mathrm{O}(t(n))$ CoT steps and any $\mathsf{TIME}(t(n))$ function within $\mathrm{O}(t(n)\log t(n))$ CoT steps. Notably, this is nearly the same as the CoT complexity of the class of *all* Transformers, which can compute any $\mathsf{TIME}(t(n))$ function within $\mathrm{O}(t(n))$ CoT steps. The logarithmic slowdown here is because the class of all Transformers can simulate an unbounded number of tapes while our single Transformer simulates only a finite number of tapes. Assuming $\mathsf{TIME}_2(t(n)) \neq \mathsf{TIME}(t(n))$, it is unlikely that the CoT complexity $\mathrm{O}(t(n)\log t(n))$ of $\Gamma$ for $\mathsf{TIME}(t(n))$ could be further improved to $\mathrm{O}(t(n))$.

## 4.3 Precision complexity

In this subsection, we analyze the precision complexity of our construction. We will show that our constructed $\Gamma$ can compute any $\mathsf{TIME}(t(n))$ function within $\mathrm{O}(\log(n + t(n)))$ bits of precision for any length-$n$ input; in particular, it can decide any P language within log-precision. Following the common practice in numerical analysis, we assume that each floating-point number has *significant* bits and *guard* bits (Goodman & Feldstein, 1977), where both significant bits and guard bits are used in arithmetic operations while guard bits are rounded off in number comparisons.

**Definition 4.6** (Precision complexity class). *For a complexity function $p(n)$, let $\mathsf{PREC}_\Gamma(p(n))$ be the class of functions that our constructed $\Gamma$ can compute using $\mathrm{O}(p(n))$ significant and guard bits.*

**Corollary 4.7** (Precision complexity). $\mathsf{TIME}(t(n)) \subseteq \mathsf{PREC}_\Gamma(\log(n+t(n)))$; $\mathsf{P} \subseteq \mathsf{PREC}_\Gamma(\log n)$.

*Proof sketch.* For any function $\varphi \in \mathsf{CoT}_\Gamma(t(n)\log t(n))$, since the prompt $\boldsymbol{\pi}_\varphi$ has length $\mathrm{O}(1)$, then by Section 3.3, the total length $I$ of the prompt, the tokenized input, and the CoT steps is

$$I = \mathrm{O}(1) + \mathrm{O}(n) + \mathrm{O}(t(n)\log t(n)) = \mathrm{O}(n + t(n)\log t(n)). \tag{29}$$

Thus, according to Section 3.4, all the intermediate results during computation are $\leq \mathrm{O}(1)$, and attention similarities have mutual differences $\geq \Omega\big(\frac{1}{I^{\Theta(1)}}\big) = \Omega\big(\frac{1}{(n+t(n)\log t(n))^{\Theta(1)}}\big)$. This implies

$$\mathsf{CoT}_\Gamma(t(n)\log t(n)) \subseteq \mathsf{PREC}_\Gamma(\log((n + t(n)\log t(n))^{\Theta(1)})) = \mathsf{PREC}_\Gamma(\log(n + t(n))). \tag{30}$$

It follows from Corollary 4.5 and Equation (30) that

$$\mathsf{TIME}(t(n)) \subseteq \mathsf{CoT}_\Gamma(t(n)\log t(n)) \subseteq \mathsf{PREC}_\Gamma(\log(n + t(n))). \tag{31}$$

In particular, we have

$$\mathsf{P} \subset \mathsf{TIME}(\mathrm{poly}(n)) \subseteq \mathsf{PREC}_\Gamma(\log(n + \mathrm{poly}(n))) = \mathsf{PREC}_\Gamma(\log n). \qquad \square$$

Notably, Corollary 4.7 shows that prompting a *single* Transformer can achieve the same precision complexity as that of the class of *all* Transformers: it is known that the class of all Transformers can compute any $\mathsf{TIME}(t(n))$ function within $\mathrm{O}(\log(n + t(n)))$ precision (Pérez et al., 2021) while we further show a single Transformer with prompting can as well. This suggests our precision complexity is presumably tight unless there are further advances in the complexity theory of Transformers.

## 5 Conclusion

In this work, we have shown that prompting is in fact Turing-complete: there exists a finite-size Transformer such that for any computable function, there exists a corresponding prompt following which the Transformer computes the function. Furthermore, we have shown that even though we use only a single finite-size Transformer, it can still achieve nearly the same complexity bounds as that of the class of all unbounded-size Transformers. Overall, our result reveals that prompting can enable a single finite-size Transformer to be efficiently universal, which establishes a theoretical underpinning for prompt engineering in practice.

### Acknowledgments

This work is supported by NSF (2416070). The content of the information in this document does not necessarily reflect the position or the policy of the Government, and no official endorsement should be inferred. The U.S. Government is authorized to reproduce and distribute reprints for Government purposes notwithstanding any copyright notation here on.

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

---

**Algorithm 1** Autoregressive generation: $\text{generate}_\Gamma(\boldsymbol{x})$

---

**Input:** decoder-only Transformer $\Gamma : \Sigma^+ \to \Sigma$; nonempty input $\boldsymbol{x} \in \Sigma^+$; stop token $\$ \in \Sigma$
**Output:** generated output $\in \Sigma^+ \cup \Sigma^\omega$
 1: **repeat**
 2:     generate next token $c \leftarrow \Gamma(\boldsymbol{x})$
 3:     append next token $\boldsymbol{x} \leftarrow \boldsymbol{x} \cdot c$
 4: **until** $c = \$$
 5: **return** generated output $\boldsymbol{x}$

---

**Algorithm 2** Extracting output from CoT (readout)

---

**Input:** generated CoT $\boldsymbol{v} \in \Sigma^+$
**Output:** extracted output $\boldsymbol{y} \in \{0, 1\}^*$
 1: initialize the left index $i \leftarrow -2$
 2: **while** $v_i \neq$ : **do**
 3:     decrement the left index $i \leftarrow i - 1$
 4: **end while**
 5: **return** extracted output $\boldsymbol{v}_{i+1:-1}$

---

## CONTENTS

## A TWO-TAPE TMS V.S. 2-PTMS

In this section, we characterize the relation between two-tape TMs and 2-PTMs.

### A.1 2-PTMS SIMULATING TWO-TAPE TMS

In this subsection, we show the correctness of the construction presented in Section 4.1.

For each non-halting state $q \in Q \setminus \{q_{\text{halt}}\}$, since we use $1+1+6+6+1+6+6 = 27$ instructions to simulate its transition rules, we put these instructions at $\boldsymbol{\iota}_{27q:27q+27}$. Thus, the instructions for state $q$ starts at $\iota_{27q}$. Besides that, for the halting state $q_{\text{halt}} = K$, we put the halting instruction after the last non-halting state $K - 1$. Thus, the halting instruction is at $\iota_{27(K-1)+27} = \iota_{27K} = \iota_{27q_{\text{halt}}}$. To recapitulate, the instructions for every state $q \in Q$ start at $\iota_{27q}$.

Hence, to make a state transition to $q' \in Q$, we should go to instruction $\iota_{27q'}$. Since we do not know pointed cell values after we move tape heads, We use two go-to instructions $\langle \mathtt{A\,!}\,_{27q'}, \mathtt{A\,?}\,_{27q'} \rangle$ with opposite conditions to ensure a state transition to $q'$. This establishes the correctness of $\boldsymbol{\eta}_{(q', c_{\mathtt{A}}, d_{\mathtt{A}}, c_{\mathtt{B}}, d_{\mathtt{B}})}$.

To simulate transition rules $\delta(q, 0, 0), \delta(q, 0, 1), \delta(q, 1, 0), \delta(q, 1, 1)$ for each non-halting state $q \in Q \setminus \{q_{\text{halt}}\}$, we first use $\text{A}?_{27q+14}$ to check the pointed cell value of tape A and then use $\text{B}?_{27q+8}$ and $\text{B}?_{27q+21}$ to check the pointed cell value of tape B, where it is easy to see that the indices $27q + 14$, $27q + 8$, and $27q + 21$ are correct.

Finally, note that the simulation is always valid because 2-PTMs have bi-directional tapes while TMs have uni-directional tapes. This concludes the correctness of the constructed 2-PTM $\iota$.

It follows that $\text{TIME}_2(t(n)) \subseteq \text{TIME}_{\text{2-PTM}}(t(n))$.

### A.2 BI-INFINITE TWO-TAPE TMs SIMULATING 2-PTMs

In this subsection, we show that any 2-PTM can be simulated by a TM over two *bi*-infinite tapes.

For any $\varphi \in \text{TIME}_{\text{2-PTM}}(t(n))$, there exists a 2-PTM $\iota$ that computes $S(\varphi(\boldsymbol{x}))$ from $S(\boldsymbol{x})$ within time $\text{O}(t(n))$. Let $K := |\iota|$. We will construct a two-tape TM $M = (Q := \{0, 1, \dots, K\}, q_{\text{start}} := 0, q_{\text{halt}} := K, \{0, 1\}, \_ := 0, m := 2, \delta : (Q \setminus \{q_{\text{halt}}\}) \times \{0, 1\}^2 \to Q \times (\{0, 1\} \times \{\text{L}, \text{S}, \text{R}\})^2)$ that simulates the 2-PTM $\iota$ within time $\text{O}(t(n))$, where we simulate each instruction $\iota_j$ $(j = 0, \dots, |\iota| - 1)$ using one state $j$. Let $c_{\text{A}}, c_{\text{B}} \in \{0, 1\}$ denote the pointed cell value on tapes A and B, respectively.

If $\iota_j = \#$, then we only make a transition to the halting state $q_{\text{halt}}$:
$$\delta(j, c_{\text{A}}, c_{\text{B}}) := (q_{\text{halt}}, c_{\text{A}}, \text{S}, c_{\text{B}}, \text{S}). \tag{32}$$
If $\iota_j = \text{A}d_{\text{A}}$ $(d_{\text{A}} \in \{\text{L}, \text{R}\})$, then we move the head for tape A and keep the head for tape B:
$$\delta(j, c_{\text{A}}, c_{\text{B}}) := (j + 1, c_{\text{A}}, d_{\text{A}}, c_{\text{B}}, \text{S}). \tag{33}$$
If $\iota_j = \text{B}d_{\text{B}}$ $(d_{\text{B}} \in \{\text{L}, \text{R}\})$, then we move the head for tape B and keep the head for tape A:
$$\delta(j, c_{\text{A}}, c_{\text{B}}) := (j + 1, c_{\text{A}}, \text{S}, c_{\text{B}}, d_{\text{B}}). \tag{34}$$
If $\iota_j = \text{A}c'_{\text{A}}$ $(c'_{\text{A}} \in \{0, 1\})$, then we write $c'_{\text{A}}$ to tape A and keep tape heads:
$$\delta(j, c_{\text{A}}, c_{\text{B}}) := (j + 1, c'_{\text{A}}, \text{S}, c_{\text{B}}, \text{S}). \tag{35}$$
If $\iota_j = \text{B}c'_{\text{B}}$ $(c'_{\text{B}} \in \{0, 1\})$, then we write $c'_{\text{B}}$ to tape B and keep tape heads:
$$\delta(j, c_{\text{A}}, c_{\text{B}}) := (j + 1, c_{\text{A}}, \text{S}, c'_{\text{B}}, \text{S}). \tag{36}$$
If $\iota_j = \tau!_k$ $(\tau \in \{\text{A}, \text{B}\})$, we go to $k$ if the pointed cell on tape $\tau$ is 0 and to $j + 1$ otherwise:
$$\delta(j, c_{\text{A}}, c_{\text{B}}) := \begin{cases} (k, c_{\text{A}}, \text{S}, c_{\text{B}}, \text{S}) & \text{if } c_\tau = 0, \\ (j + 1, c_{\text{A}}, \text{S}, c_{\text{B}}, \text{S}) & \text{if } c_\tau \neq 0. \end{cases} \tag{37}$$
If $\iota_j = \tau?_k$ $(\tau \in \{\text{A}, \text{B}\})$, we go to $k$ if the pointed cell on tape $\tau$ is 1 and to $j + 1$ otherwise:
$$\delta(j, c_{\text{A}}, c_{\text{B}}) := \begin{cases} (k, c_{\text{A}}, \text{S}, c_{\text{B}}, \text{S}) & \text{if } c_\tau = 1, \\ (j + 1, c_{\text{A}}, \text{S}, c_{\text{B}}, \text{S}) & \text{if } c_\tau \neq 1. \end{cases} \tag{38}$$
The correctness of the construction above is clear. Since $M$ simulates each $\iota_j$ through $1 = \text{O}(1)$ transition, then $M$ also has time complexity $\text{O}(t(n))$.

## B PROOF OF THEOREM 3.1

In this section, we present our detail construction of the Transformer $\Gamma$ as the proof of Theorem 3.1. We will show that we can execute the constructed prompts using affine maps, ReLU activation, layer normalization, and causal attention. Some key ideas are presented in Section 3.4 in the main paper. Unless otherwise specified, the similarity map we use in causal attention is the identity function.

### B.1 EMBEDDING LAYER

Here, we present our construction of the token embedding and the positional encoding in the embedding layer $\Gamma_{\text{emb}}$. Let $\boldsymbol{v} = v_0 \cdots v_{|\boldsymbol{v}|-1} \in \Sigma^+$ denote the input token sequence of the Transformer. For each token $v_i$ $(i = 0, \dots, v_{|\boldsymbol{v}|-1})$, we use the one-hot representation as its embedding:
$$z_i^{\text{is } \sigma} := 1_{[v_i = \sigma]}, \qquad \sigma \in \Sigma. \tag{39}$$
As described in Section 3.4, we use the following positional encoding for each position $i$:
$$p_i := 1 - \frac{(i + 1)(i + 2) + 1}{\sqrt{(i + 1)^2 + 1}\sqrt{(i + 2)^2 + 1}}. \tag{40}$$

## B.2 OTHER POSITIONAL IDENTIFIERS

In addition to the positional encoding, we compute three other positional identifiers to be used later. First, we use causal attention with query 1, key 1, and value $z_j^{\text{is}\,\hat{}}$ to compute

$$z_i^{\text{pos, 1}} := \frac{1}{i+1} \sum_{j=0}^{i} z_j^{\text{is}\,\hat{}} = \frac{1}{i+1}. \tag{41}$$

Next, we use layer normalization (LN) to compute two other positional identifiers:

$$(z_i^{\text{pos, 2}}, z_i^{\text{pos, 3}}) := \text{LN}(1, z_i^{\text{pos, 1}}) = \left( \frac{i+1}{\sqrt{(i+1)^2+1}}, \frac{1}{\sqrt{(i+1)^2+1}} \right). \tag{42}$$

## B.3 PARSING CoT STEPS

Since the prompt and the CoT share some tokens, we need distinguish CoT steps from the prompt according to the position of the delimiter token $. Specifically, we need an indicator of whether each token $v_i$ is located after the delimiter token $ or not. Thus, we use causal attention with query 1, key 1, and value $z^{\text{is}\,\$}$ to compute the discounted indicator:

$$z_i^{\text{after delim, disc}} := \frac{1}{i+1} \sum_{j=0}^{i} z_j^{\text{is}\,\$} = \frac{1_{[\exists j \leq i: v_j = \$]}}{i+1}. \tag{43}$$

Then, we use layer normalization to convert it to a Boolean value:

$$z_i^{\text{after delim}} := \text{LN}(z_i^{\text{after delim, disc}}) = 1_{[\exists j \leq i: v_j = \$]}. \tag{44}$$

Next, we check if a token is a CoT step of writing a cell using ReLU-implemented Boolean algebra:

$$z_i^{\text{is}\,\tau\,\text{write}} := \text{ReLU}(z_i^{\text{is}\,\tau 0} + z_i^{\text{is}\,\tau 1} + z_i^{\text{after delim}} - 1), \quad \tau \in \{\text{A}, \text{B}\}; \tag{45}$$

we can compute the value that is being written using ReLU:

$$z_i^{\tau\,\text{write}} := \text{ReLU}(z_i^{\text{is}\,\tau 1} + z_i^{\text{after delim}} - 1), \qquad \tau \in \{\text{A}, \text{B}\}; \tag{46}$$

we can also compute the direction of head move in a CoT step using ReLU:

$$z_i^{\tau\,\text{move}} := \text{ReLU}(z_i^{\text{is}\,\tau\text{R}} + z_i^{\text{after delim}} - 1) - \text{ReLU}(z_i^{\text{is}\,\tau\text{L}} + z_i^{\text{after delim}} - 1), \quad \tau \in \{\text{A}, \text{B}\}. \tag{47}$$

## B.4 RETRIEVING POINTED TAPE CELLS

Before retrieving the pointed tape cells, we need to compute the current positions of tape heads. Since attention cannot compute sums directly, we use the idea presented in Section 3.4 to compute a normalized representation of the position. Specifically, we first use causal attention with query 1, key 1, and value $z_j^{\tau\,\text{move}}$ to compute discounted head positions:

$$z_i^{\tau\,\text{cur disc}} := \frac{1}{i+1} \sum_{j=0}^{i} z_j^{\tau\,\text{move}}, \qquad \tau \in \{\text{A}, \text{B}\}. \tag{48}$$

Then, we use $z_i^{\text{pos, 1}} = \frac{1}{i+1}$ and LN to compute a normalized representation of head positions:

$$(z_i^{\tau\,\text{cur norm}}, z_i^{\tau\,\text{cur one norm}}) := \text{LN}(z_i^{\tau\,\text{cur disc}}, z_i^{\text{pos, 1}}) \tag{49}$$

$$= \left( \frac{\sum_{j=0}^{i} z_j^{\tau\,\text{move}}}{\sqrt{(\sum_{j=0}^{i} z_j^{\tau\,\text{move}})^2 + 1}}, \frac{1}{\sqrt{(\sum_{j=0}^{i} z_j^{\tau\,\text{move}})^2 + 1}} \right), \quad \tau \in \{\text{A}, \text{B}\}. \tag{50}$$

Next, we can retrieve the pointed tape cells by finding the last write step at the same position. We use causal attention with query $(1, z_i^{\tau\,\text{cur norm}}, z_i^{\tau\,\text{cur one norm}}, p_i)^{\mathsf{T}}$, key $(z_j^{\text{is}\,\tau\,\text{write}}, z_j^{\tau\,\text{cur norm}}, z_j^{\tau\,\text{cur one norm}}, -z_j^{\text{pos, 1}})^{\mathsf{T}}$, and value $(z_j^{\tau\,\text{write}}, z_j^{\tau\,\text{cur norm}}, z_j^{\text{is}\,\tau\,\text{write}})^{\mathsf{T}}$ to compute

the retrieved tape cells $(z_i^{\tau \text{ retr}}, z_i^{\tau \text{ retr cur norm}}, z_i^{\tau \text{ retr is write}})^\mathsf{T}$ ($\tau \in \{\mathtt{A}, \mathtt{B}\}$). Note that we also compute $z_i^{\tau \text{ retr cur norm}}$ and $z_i^{\tau \text{ retr is write}}$ here because there is a caveat: the pointed tape cells might not have been written yet. We use LN and ReLU to compute an indicator of whether the pointed tape cells have not been written:

$$
\begin{aligned}
z_i^{\tau \text{ not found}} &:= \text{ReLU}(\text{LN}(z_i^{\tau \text{ cur norm}} - z_i^{\tau \text{ retr cur norm}})) + \text{ReLU}(\text{LN}(z_i^{\tau \text{ retr cur norm}} - z_i^{\tau \text{ cur norm}})) \\
&= \mathbf{1}_{[z_i^{\tau \text{ cur norm}} \neq z_i^{\tau \text{ retr cur norm}}]}, \qquad\qquad\qquad\qquad\qquad\qquad \tau \in \{\mathtt{A}, \mathtt{B}\}. \quad (51)
\end{aligned}
$$

If a retreived tape cell has not been written yet, it must have a blank value 0:

$$
z_i^{\tau \text{ val}} := \text{ReLU}(z_i^{\tau \text{ retr}} - z_i^{\tau \text{ not found}} + z_i^{\tau \text{ retr is write}} - 1), \quad \tau \in \{\mathtt{A}, \mathtt{B}\}. \tag{52}
$$

## B.5 Parsing instructions in prompt

Recall that each instruction is encoded into one or more tokens. Thus, we first compute whether each token is the start of a instruction in the prompt:

$$
\begin{aligned}
z_i^{\text{is inst}} := \text{ReLU}(&z_i^{\text{is \#}} + z_i^{\text{is AL}} + z_i^{\text{is BL}} + z_i^{\text{is AR}} + z_i^{\text{is BR}} && (53) \\
&+ z_i^{\text{is A0}} + z_i^{\text{is B0}} + z_i^{\text{is A1}} + z_i^{\text{is B1}} && (54) \\
&+ z_i^{\text{is A!}} + z_i^{\text{is B!}} + z_i^{\text{is A?}} + z_i^{\text{is B?}} - z_i^{\text{after delim}}). && (55)
\end{aligned}
$$

We compute a *program index* $t$ for the start token of each instruction. We use causal attention with query 1, key 1, and value $z_j^{\text{is inst}}$ to compute the program index of each token in the prompt:

$$
z_i^{\text{prog idx disc, raw}} := \frac{1}{i+1} \sum_{j=0}^{i} z_j^{\text{is inst}}. \tag{56}
$$

We further subtract 1 from $\sum_{j=0}^{i} z_j^{\text{is inst}}$ to handle the lag between the prompt and the CoT:

$$
z_i^{\text{prog idx disc}} := z_i^{\text{prog idx disc, raw}} - z^{\text{pos, 1}} = \frac{1}{i+1} \sum_{j=0}^{i} z_j^{\text{is inst}} - \frac{1}{i+1}. \tag{57}
$$

Then, we use LN to compute a normalized representation of $\sum_{j=0}^{i} z_j^{\text{is inst}} - 1$:

$$
(z_i^{\text{prog idx norm}}, z_i^{\text{prog idx one norm}}) := \text{LN}(z_i^{\text{prog idx disc}}, z_i^{\text{pos, 1}}) \tag{58}
$$

$$
= \left( \frac{\sum_{j=0}^{i} z_j^{\text{is inst}} - 1}{\sqrt{(\sum_{j=0}^{i} z_j^{\text{is inst}} - 1)^2 + 1}}, \frac{1}{\sqrt{(\sum_{j=0}^{i} z_j^{\text{is inst}} - 1)^2 + 1}} \right). \tag{59}
$$

Besides that, since go-to's are special instructions that need multiple tokens to encode, we also check whether a token is the start of a go-to instruction:

$$
z_i^{\text{is goto cond}} := z_i^{\text{is A!}} + z_i^{\text{is B!}} + z_i^{\text{is A?}} + z_i^{\text{is B?}}. \tag{60}
$$

We also need a *go-to index* for tokens $-, +, \mathtt{@}$ in the prompt to mark the order of tokens within each go-to instruction. To compute it, we first compute the reciprocal number of tokens between $v_i$ and the start token $v_{i'}$ of the current go-to instruction, using causal attention with query 1, key $z_j^{\text{prog idx norm}}$, and value $z_j^{\text{is goto cond}}$:

$$
z_i^{\text{goto one disc}} := \frac{\sum_{j=i'}^{i} z_j^{\text{is goto cond}}}{\sum_{j=i'}^{i} 1} = \frac{1}{i - i' + 1}. \tag{61}
$$

Then, we use LN to compute a normalized representation of the go-to index:

$$
(z_i^{\text{goto idx norm, raw}}, z_i^{\text{goto one norm, raw}}) := \text{LN}(1 - z_i^{\text{goto one disc}}, z_i^{\text{goto one disc}}) \tag{62}
$$

$$
= \left( \frac{i - i' + 1}{\sqrt{(i - i' + 1)^2 + 1}}, \frac{1}{\sqrt{(i - i' + 1)^2 + 1}} \right). \tag{63}
$$

To match the go-to index of non-go-to tokens, we finally adjust them if $v_i$ is the start token $v_i'$ of the current go-to instruction:

$$z_i^{\text{goto idx norm}} := z_i^{\text{goto idx norm, raw}} + z_i^{\text{is goto cond}}, \tag{64}$$

$$z_i^{\text{goto one norm}} := z_i^{\text{goto one norm, raw}} - z_i^{\text{is goto cond}}. \tag{65}$$

That is, when $v_i$ is the start token $v_i'$ of a go-to instruction, we instead have

$$(z_i^{\text{goto idx norm}}, z_i^{\text{goto one norm}}) = (1, 0). \tag{66}$$

## B.6 LOCATING CURRENT CoT STEP

Here, we describe how to locate the current instruction to execute. We can imagine a program pointer $t$ that indicates the instruction $\iota_t$ we are currently executing. A caveat here is that each go-to instruction needs multiple CoT steps. Thus, it is important to check whether it is during a go-to instruction or not.

First, we compute how each go-to step contributes to the program pointer via ReLU:

$$z_i^{\text{prog move}} := \text{ReLU}(z_i^{\text{is +}} + z_i^{\text{after delim}} - 1) - \text{ReLU}(z_i^{\text{is -}} + z_i^{\text{after delim}} - 1). \tag{67}$$

Then, we use causal attention with query 1, key 1, and value $z_i^{\text{prog move}}$ to compute the discounted total contribution:

$$z_i^{\text{prog move disc}} := \frac{1}{i+1} \sum_{j=0}^{i} z_j^{\text{prog move}}; \tag{68}$$

and use LN to compute a normalized representation of $\sum_{j=0}^{i} z_j^{\text{prog move}}$:

$$(z_i^{\text{prog move norm}}, z_i^{\text{prog move one norm}}) := \text{LN}(z_i^{\text{prog move disc}}, z_i^{\text{pos, 1}}) \tag{69}$$

$$= \left( \frac{\sum_{j=0}^{i} z_j^{\text{prog move}}}{\sqrt{(\sum_{j=0}^{i} z_j^{\text{prog move}})^2 + 1}}, \frac{1}{\sqrt{(\sum_{j=0}^{i} z_j^{\text{prog move}})^2 + 1}} \right). \tag{70}$$

Next, we check whether the token $v_i$ is the start token $v_i'$ of the current execution step record in the CoT:

$$z_i^{\text{is rec start}} := z_i^{\text{is ^}} + \text{ReLU}(z_i^{\text{is AL}} + z_i^{\text{is BL}} + z_i^{\text{is AR}} + z_i^{\text{is BR}} \tag{71}$$

$$+ z_i^{\text{is A0}} + z_i^{\text{is B0}} + z_i^{\text{is A1}} + z_i^{\text{is B1}} \tag{72}$$

$$+ z_i^{\text{is /}} + z_i^{\text{is =}} + z_i^{\text{after delim}} - 1), \tag{73}$$

where we also add $z_i^{\text{is ^}}$ here for convenience later. Similarly, we check whether the token $v_i$ is the ending token $v_i'$ of the current execution step record in the CoT:

$$z_i^{\text{is rec end}} := \text{ReLU}(z_i^{\text{is \$}} + z_i^{\text{is AL}} + z_i^{\text{is BL}} + z_i^{\text{is AR}} + z_i^{\text{is BR}} \tag{74}$$

$$+ z_i^{\text{is A0}} + z_i^{\text{is B0}} + z_i^{\text{is A1}} + z_i^{\text{is B1}} \tag{75}$$

$$+ z_i^{\text{is @}} + z_i^{\text{after delim}} - 1). \tag{76}$$

Then, imagine that each token has a *record index* to mark which each execution step it belongs to. To compute it, we first compute the reciprocal number of execution steps $t'$ so far, using causal attention with query 1, key $z_j^{\text{is rec start}}$, and value $z_j^{\text{is ^}}$:

$$z_i^{\text{prog rec one disc}} := \frac{\sum_{j=0}^{i} z_j^{\text{is rec start}} z_j^{\text{is ^}}}{\sum_{j=0}^{i} z_j^{\text{is rec start}}} = \frac{1}{t'+1}. \tag{77}$$

We then compute a normalized representation of $t'+1$ using LN:

$$(z_i^{\text{prog rec norm}}, z_i^{\text{prog rec one norm}}) := \text{LN}(1, z_i^{\text{prog rec one disc}}) \tag{78}$$

$$= \left( \frac{t'+1}{\sqrt{(t'+1)^2 + 1}}, \frac{1}{\sqrt{(t'+1)^2 + 1}} \right). \tag{79}$$

Next, since each go-to execution step needs multiple CoT steps, we need to handle go-to execution step records specially when the go-to condition is satisfied. We can image that each CoT step has a *temporary program index* that marks the order of each token in a go-to execution step record. To compute it, we first compute the reciprocal number of tokens between $v_i$ and the start token $v_{i'}$ of the current go-to execution step record, using causal attention with query 1, key $-z_j^{\text{prog rec one disc}}$, and value $z_j^{\text{is }=}$:

$$z_i^{\text{prog tmp one disc}} := \frac{\sum_{j=i'}^{i} z_j^{\text{is }=}}{\sum_{j=i'}^{i} 1} = \frac{1}{i - i' + 1}. \tag{80}$$

With this, we can compute a normalized representation of $i - i' + 1$ using LN:

$$(z_i^{\text{prog tmp norm, raw}}, z_i^{\text{prog tmp one norm, raw}}) := \text{LN}(1, z_i^{\text{prog tmp one disc}}) \tag{81}$$

$$= \left( \frac{i - i' + 1}{\sqrt{(i - i' + 1)^2 + 1}}, \frac{1}{\sqrt{(i - i' + 1)^2 + 1}} \right). \tag{82}$$

To match the temporary program index of non-go-to execution steps, we further adjust them if $v_i$ is the ending token of the current go-to instruction:

$$z_i^{\text{prog tmp norm}} := \text{ReLU}(z_i^{\text{prog tmp norm, raw}} - z_i^{\text{is rec end}}) + z_i^{\text{is rec end}}, \tag{83}$$

$$z_i^{\text{prog tmp one norm}} := \text{ReLU}(z_i^{\text{prog tmp one norm, raw}} - z_i^{\text{is rec end}}). \tag{84}$$

That is, when $v_i$ is the ending token of the current go-to instruction, we instead have

$$(z_i^{\text{prog tmp norm}}, z_i^{\text{prog tmp one norm}}) = (1, 0). \tag{85}$$

We will use the quantities above to identify the instruction to execute.

## B.7 Executing next CoT step

To generate the next token, we need to execute the current instruction and record it via a CoT step. Before that, we check whether $v_i$ belongs to a satisfied go-to execution step record:

$$z_i^{\text{is rec goto}} := \text{ReLU}(z_i^{\text{is }=} + z_i^{\text{is }-} + z_i^{\text{is }+} + z_i^{\text{after delim}} - 1). \tag{86}$$

Another quantity we will need is how each CoT step contributes to the current program pointer:

$$z_i^{\text{prog cur move}} := \text{ReLU}(z_i^{\text{is AL}} + z_i^{\text{is BL}} + z_i^{\text{is AR}} + z_i^{\text{is BR}} \tag{87}$$

$$+ z_i^{\text{is A0}} + z_i^{\text{is B0}} + z_i^{\text{is A1}} + z_i^{\text{is B1}} \tag{88}$$

$$+ z_i^{\text{is }/} + z_i^{\text{is }+} + z_i^{\text{after delim}} - 1) - \text{ReLU}(z_i^{\text{is }-} + z_i^{\text{after delim}} - 1). \tag{89}$$

We also compute an auxiliary bound $z_i^{\text{prog rec diff}}$ on the differences between some attention scores using causal attention with query $(z_i^{\text{prog rec norm}}, z_i^{\text{prog rec one norm}})^{\mathsf{T}}$, key $(z_j^{\text{pos, 2}}, z_j^{\text{pos, 3}})^{\mathsf{T}}$, and value $p_j$. We further compute an auxiliary quantity:

$$z_i^{\text{prog rec bias}} := 3 - \frac{1}{2} \text{ReLU}(z_i^{\text{prog rec diff}} + 2z_i^{\text{is rec goto}} - 2). \tag{90}$$

It is easy to see that $z_i^{\text{prog rec diff}} < 3$ if and only if $z_i^{\text{is rec goto}} = 1$. Using this quantity, we will be able to avoid attending to tokens outside the current execution step record. Next, we use it to identify the next instruction to execute. We first compute the discounted program index of the instruction that we need to execute, which is proportional to the sum of $z_j^{\text{prog cur move}}$ until the beginning token $v_{i'}$ of the current execution step record, using causal attention with query $(z_i^{\text{prog rec bias}}, -z_i^{\text{prog rec norm}}, -z_i^{\text{prog rec one norm}})^{\mathsf{T}}$, key $(1, z_j^{\text{prog rec norm}}, z_j^{\text{prog rec one norm}})^{\mathsf{T}}$, value $(z_j^{\text{prog cur move}}, z_j^{\text{is }\hat{}})^{\mathsf{T}}$, and similarity function $\text{sim}(x) := 2 - \text{ReLU}(2 - x) = \min\{x, 2\}$:

$$z_i^{\text{prog cur disc}} := \frac{\sum_{j=0}^{i'} z_j^{\text{prog cur move}}}{\sum_{j=0}^{i'} 1} = \frac{\sum_{j=0}^{i'} z_j^{\text{prog cur move}}}{i' + 1}, \tag{91}$$

$$z_i^{\text{prog cur one disc}} := \frac{\sum_{j=0}^{i'} z_j^{\text{is }\hat{}}}{\sum_{j=0}^{i'} 1} = \frac{1}{i' + 1}. \tag{92}$$

Then, we use LN to compute a normalized representation of $\sum_{j=0}^{i'} z_j^{\text{prog cur move}}$:

$$(z_i^{\text{prog cur norm}}, z_i^{\text{prog cur one norm}}) := \text{LN}(z_i^{\text{prog cur disc}}, z_i^{\text{prog cur one disc}}) \tag{93}$$

$$= \left( \frac{\sum_{j=0}^{i} z_j^{\text{prog cur move}}}{\sqrt{(\sum_{j=0}^{i} z_j^{\text{prog cur move}})^2 + 1}}, \frac{1}{\sqrt{(\sum_{j=0}^{i} z_j^{\text{prog cur move}})^2 + 1}} \right). \tag{94}$$

Next, we find the instruction $\iota_t$ whose program index matches the current program pointer and whose go-to index matches the current temporary program index, using causal attention with query $(z_i^{\text{prog cur norm}}, z_i^{\text{prog cur one norm}}, z_i^{\text{prog tmp norm}}, z_i^{\text{prog tmp one norm}}, p_i, -1)^{\mathsf{T}}$, key $(z_j^{\text{prog idx norm}}, z_j^{\text{prog idx one norm}}, z_j^{\text{goto idx norm}}, z_j^{\text{goto one norm}}, z_j^{\text{pos, 1}}, 1)^{\mathsf{T}}$, and value $z_j^{\text{is } \sigma}$:

$$z_i^{\text{token is } \sigma} := z_t^{\text{is } \sigma}, \quad \sigma \in \{\texttt{\#}, \texttt{AL}, \texttt{BL}, \texttt{AR}, \texttt{BR}, \texttt{A0}, \texttt{B0}, \texttt{A1}, \texttt{B1}, \texttt{A!}, \texttt{B!}, \texttt{A?}, \texttt{B?}, \texttt{-}, \texttt{+}, \texttt{@}\}; \tag{95}$$

$$z_i^{\text{token is :}} := z_t^{\text{is \#}}. \tag{96}$$

Note that exactly one of them is $1$. It remains to execute the instruction $\iota_t$ and record its outcome. If $z_i^{\text{token is } \sigma} = 1$ for some $\sigma \in \{\texttt{A!}, \texttt{B!}, \texttt{A?}, \texttt{B?}\}$, then we need to check whether the go-to condition is satisfied or not:

$$z_i^{\text{sat A!}} := \text{ReLU}(z_i^{\text{token is A!}} - z_i^{\text{A val}}), \tag{97}$$

$$z_i^{\text{sat B!}} := \text{ReLU}(z_i^{\text{token is B!}} - z_i^{\text{B val}}), \tag{98}$$

$$z_i^{\text{sat A?}} := \text{ReLU}(z_i^{\text{token is A?}} + z_i^{\text{A val}} - 1), \tag{99}$$

$$z_i^{\text{sat B?}} := \text{ReLU}(z_i^{\text{token is B?}} + z_i^{\text{B val}} - 1). \tag{100}$$

They enable us to choose $=$ (satisfied) or $/$ (unsatisfied):

$$z_i^{\text{token is =}} := z_i^{\text{sat A!}} + z_i^{\text{sat B!}} + z_i^{\text{sat A?}} + z_i^{\text{sat B?}}, \tag{101}$$

$$z_i^{\text{token is /}} := z_i^{\text{token is A!}} + z_i^{\text{token is B!}} + z_i^{\text{token is A?}} + z_i^{\text{token is B?}} - z_i^{\text{token is =}}. \tag{102}$$

## B.8 Extracting final output

After execution, the content left on the tape $\texttt{A}$ is the Shannon-encoded output $S(\varphi(\boldsymbol{x}))$. We need to extract the decoded output $\varphi(\boldsymbol{x})$ from the previous CoT steps.

First, we need to check whether we have arrived at the final output stage:

$$z_i^{\text{is read key}} := z_i^{\text{is :}} + z_i^{\text{is 0}} + z_i^{\text{is 1}}. \tag{103}$$

Suppose that we are now trying to find the $k$-th token of the output $\varphi(\boldsymbol{x})$. Recall that the $k$-th token ($k \geq 0$) of the output $\varphi(\boldsymbol{x})$ corresponds to the $(2k)$-th and $(2k+1)$-th tokens of $S(\varphi(\boldsymbol{x}))$. To help compute the indices $2k$ and $2k + 1$, we compute the following shifts:

$$z_i^{\text{read cur0 shift}} := 2(z_i^{\text{is 0}} + z_i^{\text{is 1}}), \tag{104}$$

$$z_i^{\text{read cur1 shift}} := z_i^{\text{is :}} + 2(z_i^{\text{is 0}} + z_i^{\text{is 1}}). \tag{105}$$

Suppose that token $:$ is the $t$-th token in the generated CoT. We use causal attention with query $1$, key $z_j^{\text{is read key}}$, and value $(z_j^{\text{read cur0 shift}}, z_j^{\text{read cur1 shift}}, z_j^{\text{is :}})^{\mathsf{T}}$ to compute the discounted $2k$ and $2k + 1$ for $i = t + k$:

$$z_i^{\text{read cur0 disc}} := \frac{1}{\sum_{j=0}^{i} z_j^{\text{is read key}}} \sum_{j=0}^{i} z_j^{\text{read cur0 shift}} = \frac{2k}{i - t + 1}, \tag{106}$$

$$z_i^{\text{read cur1 disc}} := \frac{1}{\sum_{j=0}^{i} z_j^{\text{is read key}}} \sum_{j=0}^{i} z_j^{\text{read cur1 shift}} = \frac{2k + 1}{i - t + 1}, \tag{107}$$

$$z_i^{\text{read one disc}} := \frac{1}{\sum_{j=0}^{i} z_j^{\text{is read key}}} \sum_{j=0}^{i} z_j^{\text{is :}} = \frac{1}{i - t + 1}. \tag{108}$$

We use layer normalization to compute representations of $2k$ and $2k + 1$:

$$(z_i^{\text{read cur0 norm}}, z_i^{\text{read cur0 one norm}}) := \text{LN}(z_i^{\text{read cur0 disc}}, z_i^{\text{read one disc}})$$
$$= \left( \frac{2k}{\sqrt{(2k)^2 + 1}}, \frac{1}{\sqrt{(2k)^2 + 1}} \right), \tag{109}$$

$$(z_i^{\text{read cur1 norm}}, z_i^{\text{read cur1 one norm}}) := \text{LN}(z_i^{\text{read cur1 disc}}, z_i^{\text{read one disc}})$$
$$= \left( \frac{2k + 1}{\sqrt{(2k + 1)^2 + 1}}, \frac{1}{\sqrt{(2k + 1)^2 + 1}} \right). \tag{110}$$

Next, similarly with Appendix B.4, we retrieve the $(2k)$-th and $(2k + 1)$-th tokens of $S(\varphi(\boldsymbol{x}))$ as follows. We use causal attention with query $(1, z_i^{\text{read cur0 norm}}, z_i^{\text{read cur0 one norm}}, p_i)^{\mathsf{T}}$, key $(z_j^{\text{is A write}}, z_j^{\text{A cur norm}}, z_j^{\text{A cur one norm}}, -z_j^{\text{pos, 1}})^{\mathsf{T}}$, and value $(z_j^{\text{A write}}, z_j^{\text{A cur norm}}, z_j^{\text{is A write}})^{\mathsf{T}}$ to compute the retrieved tape cell status $(z_i^{\text{read retr0}}, z_i^{\text{read retr0 cur norm}}, z_i^{\text{read retr0 is write}})^{\mathsf{T}}$. We use causal attention with query $(1, z_i^{\text{read cur1 norm}}, z_i^{\text{read cur1 one norm}}, p_i)^{\mathsf{T}}$, key $(z_j^{\text{is A write}}, z_j^{\text{A cur norm}}, z_j^{\text{A cur one norm}}, -z_j^{\text{pos, 1}})^{\mathsf{T}}$, and value $(z_j^{\text{A write}}, z_j^{\text{A cur norm}}, z_j^{\text{is A write}})^{\mathsf{T}}$ to compute the retrieved tape cell status $(z_i^{\text{read retr1}}, z_i^{\text{read retr1 cur norm}}, z_i^{\text{read retr1 is write}})^{\mathsf{T}}$. We use LN and ReLU to compute an indicator of whether the retrieved tape cells have not been written:

$$z_i^{\text{read retr0 not found}} := \text{ReLU}(\text{LN}(z_i^{\text{read cur0 norm}} - z_i^{\text{read retr0 cur norm}})) + \text{ReLU}(\text{LN}(z_i^{\text{read retr0 cur norm}} - z_i^{\text{read cur0 norm}}))$$
$$= 1_{[z_i^{\text{read cur0 norm}} \neq z_i^{\text{read retr0 cur norm}}]}, \tag{111}$$

$$z_i^{\text{read retr1 not found}} := \text{ReLU}(\text{LN}(z_i^{\text{read cur1 norm}} - z_i^{\text{read retr1 cur norm}})) + \text{ReLU}(\text{LN}(z_i^{\text{read retr1 cur norm}} - z_i^{\text{read cur1 norm}}))$$
$$= 1_{[z_i^{\text{read cur1 norm}} \neq z_i^{\text{read retr1 cur norm}}]}. \tag{112}$$

Finally, we can obtain the values of cells $2k$ and $2k + 1$:

$$z_i^{\text{read retr0 val}} := \text{ReLU}(z_i^{\text{read retr0}} - z_i^{\text{read retr0 not found}} + z_i^{\text{read retr0 is write}} - 1), \tag{113}$$

$$z_i^{\text{read retr1 val}} := \text{ReLU}(z_i^{\text{read retr1}} - z_i^{\text{read retr1 not found}} + z_i^{\text{read retr1 is write}} - 1). \tag{114}$$

Using the quantities above, we can decide the next token to generate:

$$z_i^{\text{token is 0}} := 2\,\text{ReLU}(z_i^{\text{is read key}} + z_i^{\text{read retr0 val}} - z_i^{\text{read retr1 val}} - 1), \tag{115}$$

$$z_i^{\text{token is 1}} := 2\,\text{ReLU}(z_i^{\text{is read key}} + z_i^{\text{read retr0 val}} + z_i^{\text{read retr1 val}} - 2), \tag{116}$$

$$z_i^{\text{token is \$}} := 2\,\text{ReLU}(z_i^{\text{is read key}} - z_i^{\text{read retr0 val}}). \tag{117}$$

Here, we use coefficient 2 to distinguish the output extraction stage from the execution stage.

### B.9    GENERATING NEXT TOKEN

The next token to be generated by the Transformer $\Gamma$ is

$$\underset{\sigma \in \{\texttt{AL,BL,AR,BR,A0,B0,A1,B1},/,=,-,+,@,:,\texttt{0,1},\$\}}{\arg\max} z_{|\boldsymbol{v}|-1}^{\text{token is } \sigma}. \tag{118}$$

The generation procedure stops upon the generation of the ending token $\$$.[3]

### B.10    COMPOSING THE TRANSFORMER

In this subsection, we describe how to compose the aforementioned operations into a Transformer.

**Embedding layer.** Recall that the construction uses 4 positional encodings $p_i, z_i^{\text{pos, 1}}, z_i^{\text{pos, 2}}, z_i^{\text{pos, 3}}$. Thus, our token embedding and positional encoding use the first $|\Sigma| + 1$ dimensions. For the token embedding emb, the first $|\Sigma|$ dimensions are the ont-hot representation of the token, and the next dimension is zero. For example, if $v_i$ is the first token in $\Sigma$, then

$$\text{emb}(v_i) = (\ \underbrace{1, 0, \ldots, 0,}_{\text{first } |\Sigma| \text{ dims: one-hot}} \quad \underbrace{0,}_{\text{next dim: zero}} \quad \underbrace{0, 0, \ldots, 0}_{\text{slots for intermediate results}} \ )^{\mathsf{T}}. \tag{119}$$

---

[3]Nevertheless, the token $\$$ in the prompt does not stop the generation procedure.

For the positional encoding $\mathrm{pos}$,

$$\mathrm{pos}(i) = (\ \underbrace{0, 0, \ldots, 0,}_{\text{first } |\Sigma| \text{ dims: zeros}} \quad \underbrace{p_i,}_{\text{next dim: pos enc}} \quad \underbrace{0, 0, \ldots, 0}_{\text{slots for intermediate results}} \ )^\mathsf{T}. \tag{120}$$

Therefore, the embedding layer for the example above is

$$\mathrm{emb}(v_i) + \mathrm{pos}(i) = (\ \underbrace{1, 0, \ldots, 0,}_{\text{first } |\Sigma| \text{ dims: one-hot}} \quad \underbrace{p_i,}_{\text{next dim: pos enc}} \quad \underbrace{0, 0, \ldots, 0}_{\text{slots for intermediate results}} \ )^\mathsf{T}. \tag{121}$$

Here, $\mathrm{emb}(v_i) + \mathrm{pos}(i)$ is indeed the standard embedding layer in LLMs.

**Intermediate results.** Recall that each Transformer layer has a residual connection after its output. Thus, we can compute each intermediate result via a Transformer layer and add it to the hidden embedding via the residual connection. For the example above, we can construct a Transformer layer that computes $z_i^{\mathrm{pos},\,1}$, and then the hidden embedding after the residual connection of this Transformer layer is

$$(\ \underbrace{1, 0, \ldots, 0,}_{\text{first } |\Sigma| \text{ dims: one-hot}} \quad p_i, z_i^{\mathrm{pos},\,1}, \quad \underbrace{0, \ldots, 0}_{\text{slots for other intermediate results}} \ )^\mathsf{T}. \tag{122}$$

### B.11 SIZE OF THE CONSTRUCTED TRANSFORMER

In this subsection, we show that the constructed Transformer $\Gamma$ has a constant size in terms of the number of its operations and the number of bits to represent its parameters.

**Number of operations.** From the construction above, we can see that the constructed Transformer $\Gamma$ does not depend on $\varphi$ or $\boldsymbol{x}$. (To compute different functions $\varphi$, we change only the prompt but do not need to change the Transformer.) It is clear to see that our constructed Transformer has a constant number of operations.

**Number of parameter bits.** From the construction above, we can see that the constructed Transformer uses only $0, \frac{1}{2}, 1, 2, 3$ as its parameters. These numbers can be exactly expressed with a constant number of bits.

## C EXPERIMENTS

We demonstrate our construction through a Python implementation. Our code and demo results are publicly available at `https://github.com/q-rz/ICLR25-prompting-theory`.

## D CONCLUDING REMARKS

**Connection with universal TMs.** Hennie & Stearns (1966) have shown that there exists a universal TM (UTM) that can simulate any Turing machine $M$ in $\mathrm{O}(t(n) \log t(n))$ steps if $M$ halts in $t(n) \geq n$ steps. Then by Pérez et al. (2019), there exists a Transformer that simulates this UTM. However, due to the complication of the UTM, it would be quite involved to explicitly construct this Transformer. Nevertheless, this UTM has inspired us to propose 2-PTMs, which enables us to explicitly construct a simpler Transformer. Furthermore, our 2-PTMs establish the first theoretical framework to formalize *prompting*. Using our 2-PTM-based framework, we believe that people will be able to generalize more results from the classic one-model-one-task paradigm to the LLM prompting paradigm.

**Discussion on CoT complexity.** A limitation of this work is that our construction uses long CoTs for hard problems with high time complexity. However, while it might be possible to slightly improve the CoT complexity, recent theoretical evidence has suggested that long CoTs are very likely to be necessary for these hard problems. For example, Merrill & Sabharwal (2024a) have shown that any Transformer with $\mathrm{O}(\log n)$ CoT steps can solve only L problems. Assuming $\mathsf{L} \neq \mathsf{NL}$, it implies that any Transformer with $\mathrm{O}(\log n)$ CoT steps cannot even solve any NL-complete problems such

as *directed graph connectivity*. An interesting future work is to show a tight lower bound of the CoT complexity.

**Discussion on hardmax.** Following prior works (e.g., Pérez et al., 2019; Hahn, 2020), this work uses $\mathrm{hardmax}$ in attention to simplify construction. However, real-world Transformers use $\mathrm{softmax}$ in attention. Using $\mathrm{hardmax}$ instead of $\mathrm{softmax}$ is a limitation of this area. We hope to address this limitation in future work.

**Discussion on context windows.** The Transformer constructed in this work uses an unbounded context window. However, practical implementations of Transformers typically use a bounded context window due to memory considerations. This is indeed a limitation of current studies on the Turing completeness of Transformers. To address this limitation, we hope to study bounded context windows in future work.

**Discussion on learnability.** Our current work focuses on expressive power rather than learnability. While we have shown the existence of a Transformer on which prompting is Turing-complete, it does not necessarily imply that a Transformer effectively learns to simulate any 2-PTMs through CoT steps. Investigating the learnability of such a Transformer is an intriguing direction for future research.

