# OpenReview forum: "Ask, and it shall be given: On the Turing completeness of prompting"
_ICLR.cc/2025/Conference — ICLR 2025 Poster_

### Official Review · Reviewer_cbcS · 2024-10-25

**Soundness:** 3
**Presentation:** 3
**Contribution:** 3
**Rating:** 6
**Confidence:** 2

**Summary:**

This paper establishes and proves that prompting is Turing-complete: there exists a finite-size Transformer such that, for any computable function, there is a corresponding prompt that enables the Transformer to compute that function.
Additionally, the paper demonstrates that prompting is efficiently universal, meaning a single finite-size Transformer can approximate the computational complexity of the entire class of unbounded-size Transformers.

To support this, the authors introduce two-tape Post–Turing machines (2-PTMs) as a basis for constructing the prompt, chain-of-thought (CoT), input tokenizer, and Transformer model. They further prove that 2-PTMs are Turing-complete and nearly as efficient as standard Turing machines (TMs). Specifically, 2-PTM is able to compute from input x（x is within the domain of \psi) to \psi(x), within at most O(t(n)logt(n)), i.e., only logarithmic slow down over TM.

These results provide a foundation for characterizing the CoT complexity and precision complexity of the constructed Transformer. Specifically, prompting a single Transformer can compute any TIME2(t(n)) function within O(t(n)) CoT steps and any TIME(t(n)) function within O(t(n)logt(n)) CoT steps. Prompting a single Transformer can achieve the same precision complexity as that of the entire class of Transformers.

**Strengths:**

The paper provides a theoretical foundation for prompt engineering. It is well-written, with a clear structure and logical flow, making it a challenging yet engaging read.

**Weaknesses:**

There are a few minor typos to address. For instance, on line 158, the key map notation is mistakenly written as the query map notation, and on line 759, the first '1' should actually be '0'.

I have a question: does it make sense to use a one-hot representation as the embedding (lines 781–782)? Since the Transformer embedding layer doesn’t use one-hot encoding, this appears to leave a gap in the proof to me.

**Questions:**

Does it make sense to use a one-hot representation as the embedding (lines 781–782)? Since the Transformer embedding layer doesn’t use one-hot encoding, this appears to leave a gap in the proof to me.

---

> ### Author Response · Authors · 2024-11-19
> **Response to Reviewer cbcS**
>
> Thank you so much for your recognition. We are grateful that you appreciate the significance of a theoretical foundation for prompt engineering and the clear writing of this paper. We provide our response below.
>
> > *W1: There are a few minor typos to address. For instance, on line 158, the key map notation is mistakenly written as the query map notation, and on line 759, the first '1' should actually be '0'.*
>
> Thank you for carefully pointing out the typos! We have fixed them in the revised paper.
>
> > *Q1: I have a question: does it make sense to use a one-hot representation as the embedding (lines 781–782)? Since the Transformer embedding layer doesn’t use one-hot encoding, this appears to leave a gap in the proof to me.*
>
> Sorry for the confusion. We are using the **standard** embedding layer in LLMs: a token empbedding $\operatorname{emb}$ with an **additive** positional encoding $\operatorname{pos}$. That is, for each token $v_i$, the embedding layer outputs $\operatorname{emb}(v_i)+\operatorname{pos}(i)$. The token embedding map $\operatorname{emb}$ is just a formal description of `torch.nn.Embedding`.
>
> To make our construction simpler and more interpretable, we use one-hot to define the token embeddings (i.e., the parameters of `torch.nn.Embedding`). To accomodate the *additive* positional encoding and later intermediate results, we **append a few extra zeros** at the end of the one-hot. Let us elaborate on the details below.
>
> Recall that our construction uses a positional encoding $p_i$. Thus, our token embedding and positional encoding use the first $|\varSigma|+1$ dimensions.
> - For the token embedding $\operatorname{emb}$, the first $|\varSigma|$ dimensions are the ont-hot representation of the token, and the next dimension is zero. For example, if $v_i$ is the first token in $\varSigma$, then $$\operatorname{emb}(v_i)=(\underbrace{1,0,\dots,0,}\_{\text{first }|\varSigma|\text{ dims: one-hot}}\quad\underbrace{0,}\_{\text{next dim: zero}}\quad\underbrace{0,0,\ldots,0}\_\text{slots for later intermediate results})^{\mathsf T}.$$
> - For the positional encoding $\operatorname{pos}$, $$\operatorname{pos}(i)=(\underbrace{0,0,\dots,0,}\_{\text{first }|\varSigma|\text{ dims: zeros}}\quad\underbrace{p_i,}\_{\text{next dim: pos enc}}\quad\underbrace{0,0,\ldots,0}\_\text{slots for later intermediate results})^{\mathsf T}.$$
>
> Therefore, the embedding layer for the example above is
> $$\operatorname{emb}(v_i)+\operatorname{pos}(i)=(\underbrace{1,0,\dots,0,}\_{\text{first }|\varSigma|\text{ dims: one-hot}}\quad\underbrace{p_i,}\_{\text{next dim: pos enc}}\quad\underbrace{0,0,\ldots,0}\_\text{slots for later intermediate results})^{\mathsf T}.$$
> Here, $\operatorname{emb}(v_i)+\operatorname{pos}(i)$ is indeed the standard embedding layer in LLMs.
>
> We have added an elaboration to Appendix B.10.

---

> > ### Comment · Reviewer_cbcS · 2024-11-26
> >
> > Thank you for the reply. I have no further questions.

---

### Official Review · Reviewer_k74c · 2024-11-04

**Soundness:** 4
**Presentation:** 3
**Contribution:** 2
**Rating:** 5
**Confidence:** 3

**Summary:**

This paper demonstrates that a family of decoder-only transformers achieves Turing-completeness through appropriate prompting. The authors establish this by constructing transformers that can simulate a two-tape Post-Turing Machine (2-PTM), a model known to be Turing-complete. Their approach involves encoding 2-PTM programs and inputs into prompts, enabling the transformer to simulate the execution trace of the 2-PTM. A subsequent readout process extracts the final output of the 2-PTM from the transformer's output. The paper also provides complexity bounds for the chain-of-thought (CoT) steps required in the simulation, proving its efficiency.

**Strengths:**

+ The paper presents a new reduction that proves Turing-completeness of transformers with hardmax attention.
+ The authors deliver a thorough construction and proof for their Turing-completeness result.

**Weaknesses:**

- The paper lacks sufficient justification for the novelty of its results. The Turing-completeness of transformers with prompting appears to follow naturally from existing work. It’s well-known that a Universal Turing Machine (UTM) can simulate any Turing machine by encoding the machine as part of the input. Given prior work [1], which shows that transformers with hard attention are Turing-complete, it’s intuitive that a family of transformers can simulate a UTM. Consequently, encoding any Turing machine within a prompt for a transformer simulating a UTM to achieve Turing-completeness appears to be a straightforward implication.
- Regarding complexity, the simulation results presented do not appear groundbreaking. Previous work [2] establishes that there exists a UTM capable of simulating any Turing machine in $O(T(n)\log(T(n)))$ steps, given the machine halts within $T(n)$ steps. Since the simulation in [1] requires only a single transformer step per Turing machine step, it follows that [1]'s construction can also simulate within $O(T(n)\log(T(n)))$ steps. The authors should clarify why their new construction is necessary and how it extends or innovates beyond these established results.

[1] Pérez, J., Barceló, P., & Marinkovic, J. (2021). Attention is turing-complete. Journal of Machine Learning Research, 22(75), 1-35.

[2] Hennie, F. C., & Stearns, R. E. (1966). Two-tape simulation of multitape Turing machines. Journal of the ACM (JACM), 13(4), 533-546.

**Questions:**

What is the advantage of using a 2-PTM instead of a standard Turing machine in [1]?

[1] Pérez, J., Barceló, P., & Marinkovic, J. (2021). Attention is turing-complete. Journal of Machine Learning Research, 22(75), 1-35.

---

> ### Author Response · Authors · 2024-11-19
> **Response to Reviewer k74c (Part 1/2)**
>
> Thank you so much for your insightful review and interesting questions.
> We provide our response as follows.
>
> > *W1: The paper lacks sufficient justification for the novelty of its results. The Turing-completeness of transformers with prompting appears to follow naturally from existing work. It’s well-known that a Universal Turing Machine (UTM) can simulate any Turing machine by encoding the machine as part of the input. Given prior work [1], which shows that transformers with hard attention are Turing-complete, it’s intuitive that a family of transformers can simulate a UTM. Consequently, encoding any Turing machine within a prompt for a transformer simulating a UTM to achieve Turing-completeness appears to be a straightforward implication.*
> >
> > *Q1: What is the advantage of using a 2-PTM instead of a standard Turing machine in [1]?*
>
> Thank you for the insightful question. We elaborate on our novelty as follows.
>
> - Firstly, our main goal and technical novelty are not just existence but an **explicit and simple construction**. We believe that an explicitly constructive proof is more meaningful because it **helps to further study the properties** of the construction. For example, if we did not provide an explicit construction, it would be more difficult to study the **precision complexity** (our Corollary 4.7). It also paves the way for future work to study other properties of the construction.
>
>   In fact, **we had tried the UTM approach** that the reviewer  mentioned, but technical complications arise when we attempted to *explicitly* construct a Transformer from the UTM. (i) The efficient UTM [2] is only described in words due to its complication, and it would be quite involved to explicitly construct that UTM. Meanwhile, explicitly constructed simple UTMs suffer from at least a polynomial slowdown (see, e.g., [NW09] for a survey). (ii) Since the encoding of TMs is also complex, it makes the explicit construction of the UTM even more complicated. Hence, we have decided not to use the UTM approach.
>
>   Nevertheless, **the UTM approach has inspired our 2-PTMs**. To address the aforementioned technical complications, we propose 2-PTMs as an alternative, which are much easier to encode and to parse than TMs. This enables us to explicitly construct a simple Transformer.
> - More importantly, our 2-PTMs not only help in this work but also **establish a theoretical framework** for tackling prompting-related theoretical analyses in future research. There is no prior theoretical work on the LLM prompting paradigm (i.e., one-model-many-tasks), and we believe this is because this area currently **lacks an appropriate framework to theoretically formalize "prompting."** Our 2-PTMs seem more suitable for formulating prompting than natural language and TMs because 2-PTMs are easier to encode and to parse.
>
>   Using our 2-PTM-based framework, we believe that people will be able to generalize more results from the classic one-model-one-task paradigm to the one-model-many-tasks paradigm (such as analyzing in-context learning). In particular, this work is an example showing how we can generalize the Turing completeness of the class of all Transformers [1] to the Turing completeness of prompting using our 2-PTM-based framework.
>
> We have added a discussion on novelty and connection to UTM to Appendix C (due to the page limit).

---

> ### Author Response · Authors · 2024-11-19
> **Response to Reviewer k74c (Part 2/2)**
>
> > *W2: Regarding complexity, the simulation results presented do not appear groundbreaking. Previous work [2] establishes that there exists a UTM capable of simulating any Turing machine in $O(T(n)\log T(n))$ steps, given the machine halts within $T(n)$ steps. Since the simulation in [1] requires only a single transformer step per Turing machine step, it follows that [1]'s construction can also simulate within $O(T(n)\log T(n))$ steps. The authors should clarify why their new construction is necessary and how it extends or innovates beyond these established results.*
>
> Thanks for raising this great question. As a single Transformer is more restricted than the class of all Transformers, it is unlikely for our CoT complexity to beat the best known CoT complexity of the class of all Transformers (i.e., [1]). Instead, as elaborated in our response to W1, we have decided to use 2-PTMs instead of a UTM because we hope to provide an **explicit and simple** construction.
>
> The primary goal of our complexity bounds is to demonstrate that our **simple construction** can still achieve the **same complexity bounds** as the class of all Transformers. This is in contrast to those explicitly constructed small UTMs, which suffer from at least a polynomial slowdown [NW09]. The complexity bounds suggest that 2-PTMs can indeed serve as a **desirable theoretical framework** to formulate "prompting," and we hope that our 2-PTM-based framework will facilitate future theoretical research on prompting.
>
> References:
> - [NW09] The complexity of small universal Turing machines: a survey. *Theoretical Computer Science, 410*(4-5), 443-450.
> - [1] Pérez et al. (2021). Attention is Turing-complete. *Journal of Machine Learning Research, 22*(75), 1-35.
> - [2] Hennie et al. (1966). Two-tape simulation of multitape Turing machines. *Journal of the ACM, 13*(4), 533-546.

---

> > ### Comment · Reviewer_k74c · 2024-11-26
> >
> > Thank you for your response. I agree that an explicit and straightforward construction could help further research on the properties of transformers. However, I believe that this paper does not yield sufficiently novel results from the construction. It would be better if the author could prove new results derived from this construction or provide an implementation on real machines to demonstrate the construction. I will maintain my current score.

---

> ### Author Response · Authors · 2024-11-28
> **Follow-up response**
>
> Thank you for your insightful follow-up suggestions. We provide our response below.
>
> > *I agree that an explicit and straightforward construction could help further research on the properties of transformers.*
>
> Thank you so much for recognizing our contribution.
>
> To clarify our contribution (simple constructive proof), we have added a discussion of this simple existence proof to Section 1.2.
>
> > *It would be better if the author could prove new results derived from this construction or provide an implementation on real machines to demonstrate the construction*
>
> Thank you for the great suggestion! Regarding new results, for example, we believe that our 2-PTM-based framework can also be used to give a simple constructive proof of the Turing completeness of other popular sequence models such as Mamba [TD23]. We choose Transformers in this work because it is the most popular architecture of LLMs.
>
> We demonstrate our construction via a Python implementation. Our code and results are available at **https://anonymous.4open.science/r/prompting-theory/main.ipynb**
>
> In the code, we demonstrate the constructed Transformer using two functions `Parity` and `Dyck` (30 examples in total). For example, suppose that we want to compute `Parity(1011110010)`. The prompt for the `Parity` function is:
> ```
> ^A!++++@ARARA?--@A1ARA1A?+++@ALALALALALA?++@#ARARA0ARA!-----------@A0ALALA?+++@A1A?++@A0ALA?-----------------@$
> ```
> The tokenization of `1011110010` is:
> ```
> ARARARARARARARARARARARARARARARARARARARARALALA1ALA1ALA1ALALA1ALALA1ALA1ALA1ALA1ALA1ALA1ALA1ALA1ALA1ALALA1ALA1ALA1=--------------------------------------------------------@
> ```
> The output (including CoT) generated by the Transformer is:
> ```
> /ARAR=--@ARAR=--@ARAR=--@ARAR=--@ARAR=--@ARAR=--@ARAR=--@ARAR=--@ARAR=--@ARAR/A1ARA1=+++@ALALAL=++@ARARA0AR/A0ALAL/A1=++@AL=-----------------@ALAL=++@ARARA0AR/A0ALAL=+++@A0AL=-----------------@ALAL=++@ARARA0AR=-----------@ALALALALAL=++@ARARA0AR=-----------@ALALALALAL=++@ARARA0AR=-----------@ALALALALAL=++@ARARA0AR/A0ALAL=+++@A0AL=-----------------@ALAL=++@ARARA0AR=-----------@ALALALALAL=++@ARARA0AR/A0ALAL=+++@A0AL=-----------------@ALAL=++@ARARA0AR=-----------@ALALALALAL=++@ARARA0AR=-----------@ALALALALAL/:1$
> ```
> We can see that the generated answer is indeed `1` (the part between `:` and `$` at the end of the output), which is the correct answer.
>
> We will add this experiment to Appendix D.
>
> Reference:
>
> - [TD23] Mamba: Linear-time sequence modeling with selective state spaces.

---

> > ### Comment · Reviewer_k74c · 2024-12-03
> >
> > Thank you for your response. I believe your implementation of this construction, along with the experiments, can greatly assist people in comprehending and experimenting with the Turing completeness of Transformers. If more tool chains can be developed to convert general Turing machines into Transformers, it may open up numerous new directions of work. Nevertheless, I believe this contribution should be emphasized in the main body of the article rather than in the appendix. This requires significant modifications to the article, which cannot be completed in a minor revision. Therefore, I decided to maintain my current score.

---

> > > ### Author Response · Authors · 2024-12-03
> > >
> > > Thank you so much for your constructive comments! We sincerely appreciate your recognition of the value of our construction.
> > >
> > > Following your suggestion, we will revise our introduction (specifically Contribution 1 and Technical Overview) to clearly emphasize that our primary contribution is the construction.

---

> ### Author Response · Authors · 2024-12-02
> **A gentle reminder**
>
> Dear Reviewer k74c,
>
> Hope this message finds you well.
>
> As **today (Dec 2) is the last day** for reviewers to post follow-up comments, we would like to confirm whether our responses have adequately addressed your concerns. Please let us know if you have any additional questions or require further clarifications—we would be very happy to answer them.
>
> Thank you for your time and feedback.

---

### Official Review · Reviewer_2gkz · 2024-11-04

**Soundness:** 3
**Presentation:** 3
**Contribution:** 2
**Rating:** 5
**Confidence:** 2

**Summary:**

This paper constructs a Transformer-like algorithm that, given an encoding of a particular two-tape Post-Turing machine, simulates the machine. The algorithm is "Transformer-like" in that it is a composition of a finite number of "layers," each of which has structure similar to that found in a Transformer (e.g., query, key, and value computations that feed into a (hard) attention mechanism). The authors conclude from this construction that "prompting is Turing-complete," i.e., that there exists a finite Transformer that can be prompted to efficiently simulate any program.

**Strengths:**

This paper is clearly written, and the technical details appear to be sound. The construction of the particular Transformer for simulating 2-PTMs is clever.

**Weaknesses:**

I am an outsider to this area, and am not confident in this evaluation. However, I found it difficult to understand the significance of this result.

We, of course, already know that it is possible to unambiguously specify computations with strings in a finite alphabet. We also know that finite machines with infinite tapes can execute these computations. In some sense the "prompting paradigm" (use a string to tell a machine what we want it to do) is also just the "programming paradigm." So the real question is whether a Transformer can execute a program provided to it in its prompt, and it seems to me that the answer hinges on how you define "Transformer." As you vary the definition, you will be setting up different puzzles for yourself, regarding how to encode the various necessary operations (e.g., in this paper, Boolean logic, equality checks, farthest retrieval) within a "Transformer." Why is this particular model of Transformers interesting?

It seems important for the model of the Transformer to reflect real-world Transformers in all essential respects; but in order to judge which respects are essential, more context is needed on what doubts or questions about (real-world) Transformers this research is meant to address. That is, a priori, why might someone *doubt* the existence of a Transformer that executes program instructions in its prompt? And does your theorem actually put those doubts to rest? For instance, if someone thinks Transformers are limited because they have finite context windows, this theorem will not dissuade them (it assumes infinite context windows). If someone thinks that somehow the "soft" nature of softmax attention limits Transformers' ability to correctly implement "hard" algorithms, this theorem will not persuade them otherwise (it assumes hard attention). And so on. What questions/doubts exactly is your theorem intending to settle? Is the model of Transformers that you propose sufficiently faithful to real Transformers to satisfyingly address those questions?

**Questions:**

I would love to better understand what about this result you think might be surprising. Before you proved the result, did you have any doubt it would hold? What aspects of Transformers did you imagine might make them less than Turing-complete, when given infinite CoT steps and an infinite context window?

---

> ### Author Response · Authors · 2024-11-19
> **Response to Reviewer 2gkz (Part 1/2)**
>
> Thank you for your time to provide a detailed review. We appreciate your unique perspective on this work, and we believe that your questions will greatly help to improve the clarity of this paper. We give a point-to-point response as follows.
>
> > *W1: I found it difficult to understand the significance of this result. In some sense the "prompting paradigm" (use a string to tell a machine what we want it to do) is also just the "programming paradigm."*
>
> We agree with the reviewer that one can indeed conceptually view prompting as programming, which is the idea this paper is trying convey. Despite the empirical success of the LLM prompting paradigm (e.g., general question answering and in-context learning), there is **no existing theoretical work on prompting** prior to our work. This is because people in practice prompt the LLMs in natural language, but **natural language is hard to analyze theoretically**. In fact, even though the Turing completeness of the class of all Transformers have already been proven in 2019 [PMB19], there is still not theoretical work on prompting so far until our work. We believe this is because this area currently **lacks an appropriate theoretical framework** to formalize what prompting is. To the best of our knowledge, our work is the first to abstract the natural language prompting into a formal language, in particular, a language that resembles a programming language.
>
> Furthermore, technical difficulty also arises in finding a suitable "programming language." We need to make a trade-off between the grammatical sophistication and the computational power of the language.
> - For example, popular programming languages like C or Python are not suitable because it would be very complicated to explicitly construct a Transformer to parse their sophisticated grammars.
> - Meanwhile, simple-grammar languages like Brainfuck are also not suitable because it is much less efficient than Turing machines.
>
> Hence, we need to design a new programming language that satisfies the desiderata above. Finally, we successfully designed 2-PTMs as our theoretical framework. Moreover, using our 2-PTM-based framework, we believe that people will be able to generalize more results from the classic one-model-one-task paradigm to the LLM prompting paradigm. In particular, this work is an example showing how we can generalize the Turing completeness of the class of all Transformers [PMB19] to the Turing completeness of prompting using our 2-PTM-based framework.
>
> > *W2: So the real question is whether a Transformer can execute a program provided to it in its prompt, and it seems to me that the answer hinges on how you define "Transformer." As you vary the definition, you will be setting up different puzzles for yourself, regarding how to encode the various necessary operations (e.g., in this paper, Boolean logic, equality checks, farthest retrieval) within a "Transformer."*
>
> We would like to clarify that we are using the standard definition of Transformers except that attention uses hardmax instead of softmax, which is the **standard abstraction of attention** in this area (see, e.g., [PMB19,Hahn20]). We agree that using hardmax could indeed be a limitation of this area. This is because there is technical difficulty in analyzing softmax. If one used softmax, then it would be technically hard to analyze the intermediate values $z$ due to the $\exp$ operation in softmax; in contrast, if one uses hardmax, then most of the intermediate values $z$ are simply Boolean values.
> Meanwhile, other components in our definition (layer normalization, causal attention, residual connection) are exactly what are **being used in almost all modern LLMs**. Thus, our Section 2.2 is basically a formal description of the original Transformer architecture [VSPUJGKP17], which is to make this paper self-contained.
>
> The operations we introduced in Section 3.4 (Boolean logic, equality checks, farthest retrieval) are just *tricks*
> we are using to constructe the Transformer. It is not hard to show that these operations can indeed be implemented by a standard Transformer thanks to *residual connections*. Hence, we are **not varying** the definition of Transformers; we are just **making use of** the definition.
>
> > *Q1: Why is this particular model of Transformers interesting? It seems important for the model of the Transformer to reflect real-world Transformers in all essential respects; but in order to judge which respects are essential, more context is needed on what doubts or questions about (real-world) Transformers this research is meant to address.*
>
> As elaborated above, we are studying the standard Transformer architecture **being used in almost all modern LLMs**. More specifically, almost all modern LLMs are based on decoder-only Transformers (i.e., causal attention instead of bidirectional attention), so we also use decoder-only Transformer in this work (see Section 2.2).

---

> ### Author Response · Authors · 2024-11-19
> **Response to Reviewer 2gkz (Part 2/2)**
>
> > *Q2: a priori, why might someone doubt the existence of a Transformer that executes program instructions in its prompt?*
>
> There are two main reasons for doubting the existance of such a Transformer.
> - First, it is known that Transformers are **not always universal**. For example, [Hahn20] shows that if CoT is not allowed, then any Transformer cannot even compute the parity function (an extremely simple function); furthermore, [MS24] shows that even when $O(\log n)$ CoT steps are allowed, Transformers still cannot solve the *directed graph connectivity* problem (assuming LOGSPACE $\neq$ NLOGSPACE).
> - Second, even if such a Transformer exists, it **might be inefficient** in executing programs. For example, if we replaced our 2-PTMs with classic DSW-PTMs, then the CoT complexity $O(t(n)\log t(n))$ would worsen to $O(t(n)^2)$.
>
> Therefore, despite the empirical success of Transformers, a theorist in this area might still doubt whether such an efficient Transformer exists. We will make these points clearer in the revised paper.
>
> > *Q3: And does your theorem actually put those doubts to rest? For instance, if someone thinks Transformers are limited because they have finite context windows, this theorem will not dissuade them (it assumes infinite context windows).*
>
> Sorry for the confusion. We are **not assuming** an infinite context window. Instead, we are **explicitly constructing** a Transformer that indeed has an infinite context window. If someone thinks Transformers do not have infinite context windows in practice, they are actually doubting *learnability* rather than existence of the Transformer.
> While learnability is beyond the scope of this work, we believe that it is a very intersting future direction.
>
> > *Q4: If someone thinks that somehow the "soft" nature of softmax attention limits Transformers' ability to correctly implement "hard" algorithms, this theorem will not persuade them otherwise (it assumes hard attention).*
>
> Thanks for pointing out this important aspect. While hardmax is a standard abstraction in this area,
> we agree with the reviewer that using hardmax instead of softmax could indeed be a limitation of this area. This is because there is technical difficulty in analyzing softmax.  If one used softmax, then it would be technically hard to analyze the intermediate values $z$ due to the $\exp$ operation in softmax; in contrast, if one uses hardmax, then most of the intermediate values $z$ are simply Boolean values.
> How to directly analyze softmax would be a very interesting future direction.
>
> > *Q4: What questions/doubts exactly is your theorem intending to settle? Is the model of Transformers that you propose sufficiently faithful to real Transformers to satisfyingly address those questions?*
>
> Thank you for raising these insightful questions. As elaborated in our response to Q2, since Transformers are not always universal, a theorist in this area might still doubt **whether there exists an efficient hardmax Transformer to simulate programs**. Our theorem mainly intends to settle the doubt.
> We agree that someone might doubt whether the hardmax assumption is realistic, and we hope we can address this technical limitation in future work. Apart from hardmax, we believe that our definition of Transformers is standard and realistic, which is being used in almost all modern LLMs.
>
> Due to the space limit, we have added a discuss on the aforementioned limitations and future work to Appendix C (due to the page limit).
>
> References:
> - [PMB19] On the Turing completeness of modern neural network architectures. *ICLR*, 2019.
> - [Hahn20] Theoretical limits of self-attention in neural sequence models. *Transactions of the ACL*, 2020.
> - [VSPUJGKP17] Attention is all you need. *NeurIPS*, 2017.

---

> > ### Comment · Reviewer_2gkz · 2024-11-26
> >
> > Thank you for your detailed response.
> >
> > > Sorry for the confusion. We are not assuming an infinite context window. Instead, we are explicitly constructing a Transformer that indeed has an infinite context window. If someone thinks Transformers do not have infinite context windows in practice, they are actually doubting learnability rather than existence of the Transformer. While learnability is beyond the scope of this work, we believe that it is a very interesting future direction.
> >
> > I think I see what you're saying, in that a finite Transformer (i.e., with finitely many weights) can be run with an infinite context window. However, in practice, I was under the impression that implementations keep key/value vectors in memory for only a finite window of previous tokens (often much smaller than the memory of the device). This would presumably be a limit to Turing-completeness. (I do think it's fine though--and expected--that your approach needs an infinite context window.)
> >
> > > There are two main reasons for doubting the existence of such a Transformer.
> >
> > Thanks for clarifying. I find the efficiency argument compelling. However, I also see in Reviewer k74c's review a straightforward argument for the existence of such an efficient Transformer: we already know that efficient UTMs exist, and that any TM can be simulated efficiently by a Transformer.
> >
> > I see that you responded to Reviewer k74c by claiming that your key aim was to provide a simple and explicit construction. That could be valuable, but if that is the contribution I feel the claims of the paper should be much more clearly scoped in the Introduction. That is, the Introduction should clarify that existence of an efficient Transformer for simulating arbitrary TMs encoded in the prompt is already a known result, but that explicitly constructing the Transformer is cumbersome, and this paper is providing a simple construction that makes it possible to study x,y,z further questions.

---

> ### Author Response · Authors · 2024-11-27
> **Follow-up response**
>
> Thank you for your insightful follow-up questions and suggestions. We provide our response below.
>
> > *Q6: I think I see what you're saying, in that a finite Transformer (i.e., with finitely many weights) can be run with an infinite context window. However, in practice, I was under the impression that implementations keep key/value vectors in memory for only a finite window of previous tokens (often much smaller than the memory of the device). This would presumably be a limit to Turing-completeness. (I do think it's fine though--and expected--that your approach needs an infinite context window.)*
>
> Thank you for your insightful question. We agree that using an infinite context window is a indeed limitation of this area, as practical implementations of LLMs only use finite context windows. We hope to find a better construction that only uses a finite context window in future work.
>
> We have added a discussion of this limitation to Appendix C.
>
> > *Q7: Thanks for clarifying. I find the efficiency argument compelling. However, I also see in Reviewer k74c's review a straightforward argument for the existence of such an efficient Transformer: we already know that efficient UTMs exist, and that any TM can be simulated efficiently by a Transformer.*
> >
> > *I see that you responded to Reviewer k74c by claiming that your key aim was to provide a simple and explicit construction. That could be valuable, but if that is the contribution I feel the claims of the paper should be much more clearly scoped in the Introduction. That is, the Introduction should clarify that existence of an efficient Transformer for simulating arbitrary TMs encoded in the prompt is already a known result, but that explicitly constructing the Transformer is cumbersome, and this paper is providing a simple construction that makes it possible to study x,y,z further questions.*
>
> Thank you for your suggestion. We agree that one can use UTM to give a simple existence proof. The existence proof via UTM is actually **not a known result**, but it should not be hard to be deduced by combining [HS66] and [PMB19].
>
> We have added a discussion of this existence proof into Section 1.2 in Introduction to clarify our contribution (a simple constructive proof).
>
> References:
>
> - [HS66] Two-tape simulation of multitape Turing machines. Journal of the ACM, 13(4):533-546, 1966.
> - [PMB19] On the Turing completeness of modern neural network architectures. ICLR, 2019.

---

### Official Review · Reviewer_h1Bs · 2024-11-05

**Soundness:** 3
**Presentation:** 3
**Contribution:** 3
**Rating:** 6
**Confidence:** 4

**Summary:**

This is a theoretical work whose main contribution is the proof that there exists a finite-sized transformer such that for any computable function $\varphi$ there is a corresponding prompt such that, when concatenated with an input $x$, the transformer generates an output which encodes the value $\varphi(x)$. To show this result, the authors use a new model of computations, called in the paper two-tape Post-Turing machines (2-PTMs), which extends Wang's basic machines and is inspired by the Hennie–Stearns theorem showing that any multi-type Turing machine (TM) can be simulated by a two-tape TM whose running time increases only by a logarithmic factor. The authors also provide bounds on the size of the output generated by the transformer, which depends on the size of the input and the time complexity of the multi-type TM computing the function.

**Strengths:**

The computational power of Neural Networks, and especially of Transformers, studied in this work, is an important and interesting topic that has been studied for some time. The authors have shown new results in this field and improved previously known achievements, in particular those of Perez et al. (ICLR 2019 and JMLR 2021) and Merrill & Sabharwal (ICLR, 2024). While Perez et al. (2021) have shown that for any computable function $\varphi$ there exists a Transformer that computes the function, the submitted work proves that there exists a single finite sized (decoder-only) Transformer that, when fed with promptes corresponding to a function, is Turing complete. Moreover, the performance of prompting the single Transformer achieves the same precision complexity and almost the same chain-of-thought (CoT) complexity as the Transformers presented in previous works, which, however, were not universal.

**Weaknesses:**

The topics discussed in the paper and the methods of proofs are not entirely new, and the main result, although it sheds new light on the issue of computability, is not groundbreaking in this field. Moreover, to claim that this work proves Turing completeness of prompting is in some sense an over interpretation of the achievements. First, although it is justified by the results of Hahn (2020) and others, a rather negative aspect of the results of this work is hidden in the fact that autoregressive generation has to process huge sequences of tokens (called in this and related works CoT). Second, to compute a function from the class TIME(t(n)), the constructed Transformer Γ needs $O(\log(n + t(n)))$ bits of precision. Thus, in this context, Γ is not universal in the sense that, for example, a Universal Turing machine is understood.

**Questions:**

Please explain the aspect of universality of your construction mentioned above.

Could you please discuss in more detail what is the size of the constructed Transformer.

L.129: you write that you refer to the time complexity as the the time complexity on a random-access
machine (RAM). But this seems to be not true. Could you please explain this?

L.146: why is the embedding function, a function from $\Sigma$ and not from $\Sigma^+$ or $\Nat \times \Sigma$?

L. 158: You call: key map qry_{l,k}. It should be: key_{l,k}

L. 335: why do you define E(0) := ALALA1 and not as E(0) := ALA0ALA1 ?

---

> ### Author Response · Authors · 2024-11-19
> **Response to Reviewer h1Bs (Part 1/2)**
>
> Thank you for your time to provide such a detailed review. We are delighted that you appreciate that the computational power of Transformers is an important research topic and recognize that the constructed single Transformer achieves the same precision complexity and almost the same CoT complexity as the class of Transformers. Besides that, we believe your questions will greatly help to improve the clarity of the paper. We answer your questions as follows.
>
> > *W0: The topics discussed in the paper and the methods of proofs are not entirely new, and the main result, although it sheds new light on the issue of computability, is not groundbreaking in this field. Moreover, to claim that this work proves Turing completeness of prompting is in some sense an over interpretation of the achievements.*
>
> We would like to clarify that a main contribution of this work is **how to theoretically formalize "prompting."** In fact, even though the Turing completeness of the class of all Transformers have already been proven in 2019 [PMB19], there is still not theoretical work on prompting so far until our work. We believe this is because this area currently **lacks an appropriate theoretical framework to formalize what "prompting" is.** To bridge this gap, we propose 2-PTMs as such a framework, which is easy to encode and to theoretically analyze. Using our 2-PTM-based framework, we believe that people will be able to generalize more results from the classic one-model-one-task paradigm to the one-model-many-tasks paradigm. In particular, this work is an example showing how we can generalize the Turing completeness of the class of all Transformers [PMB19] to the Turing completeness of prompting using our 2-PTM-based framework.
>
> > *W1: First, although it is justified by the results of Hahn (2020) and others, a rather negative aspect of the results of this work is hidden in the fact that autoregressive generation has to process huge sequences of tokens (called in this and related works CoT).*
>
> Thank you for raising this important point. First, we agree with the reviewer that a potential limitation is that our construction uses long CoT for hard problems with high time complexity. However, while it might be possible to slightly improve the CoT complexity, recent theoretical evidence has suggested that **long CoT is very likely to be necessary** for these hard problems. For example, [MS24] has shown that any Transformer with $O(\log n)$ CoT steps can solve **only LOGSPACE problems**. Assuming LOGSPACE $\ne$ NLOGSPACE, it implies that any Transformer with $O(\log n)$ CoT steps **cannot even solve** any NLOGSPACE-complete problems such as **directed graph connectivity**. Therefore, we believe that long CoT is very likely to be necessary for hard problems. We have added an elaboration to Appendix C (due to the page limit).
>
> > *W2: Second, to compute a function from the class $TIME(t(n))$, the constructed Transformer $\varGamma$ needs $O(\log(n+t(n)))$ bits of precision. Thus, in this context, $\varGamma$ is not universal in the sense that, for example, a Universal Turing machine is understood.*
>
> Sorry for the confusion. Our precision complexity result is not about our Turing completeness result. Let us elaborate as follows.
> - Regarding the Turing completeness, we follow prior work (e.g., [PMB19]) to assume that arithmetic computations are exact (i.e., infinite precision). This is a common assumption in the study of Turing completeness of neural sequence models (e.g., [SS92]) and, more generally, a common assumption in the theoretical study of numerical algorithms (e.g., real Turing machines [BSS89,CKKLW95]). To clarify this, we have added a footnote to the statement of Theorem 3.1.
> - The precision complexity of the Transformer is not about Turing completeness. It can be regarded as **an analog of the space complexity** of Turing machines --- even though a Turing machine is assumed to have **inifite tapes**, researchers are still interested in **how much space** is needed for a Turing machine to function. Similarly here, even though the Transformer is assumed to have infinite precision, we are still interested in how much precision is needed for the Transformer to function.

---

> > ### Author Response · Authors · 2024-11-19
> > **Response to Reviewer h1Bs (Part 2/2)**
> >
> > > *Q1: Could you please discuss in more detail what is the size of the constructed Transformer?*
> >
> > The Transformer is explicitly constructed in Appendix B. The size of a Transformer is the number of its operations and the number of bits to represent its parameters.
> > - Number of operations: Recall that our constructed Transformer is independent of the function $\varphi$ or the input $\boldsymbol x$. (To compute different functions $\varphi$, we change only the prompt but do not need to change the Transformer.) It is clear to see that our constructed Transformer has a constant number of operations.
> > - Number of parameter bits: Recall that our constructed Transformer uses only 0, 1/2, 1, 2, 3 as its parameters. These numbers can be exactly expressed with a constant number of bits.
> >
> > We have added a discussion on the size of the Transformer to Appendix B.11.
> >
> > > *Q2: L.129: you write that you refer to the time complexity as the the time complexity on a random-access machine (RAM). But this seems to be not true. Could you please explain this?*
> >
> > Sorry for the confusion. What we mean is that $\operatorname{tokenize}$ and $\operatorname{readout}$ runs in linear time on a RAM. The complexity class $\mathsf{TIME}(t(n))$ and the CoT complexity are w.r.t.\ the time complexity on a Turing machine. To avoid the confusion, we have removed that sentence and instead clarified this in the statement of Theorem 3.1.
> >
> > > *Q3: L.146: why is the embedding function, a function from $\varSigma$ and not from $\varSigma^+$ or $\mathbb N\times\varSigma$?*
> >
> > Sorry for the confusion. We are using the standard embedding layer in LLMs: a token empbedding $\operatorname{emb}$ with an **additive** positional encoding $\operatorname{pos}$. That is, for each token $v_i$, the embedding layer outputs $\operatorname{emb}(v_i)+\operatorname{pos}(i)$. Hence, you may regard the embedding layer $\operatorname{emb}+\operatorname{pos}$ as a function from $\varSigma\times\mathbb N$ while $\operatorname{emb}$ is just a function from $\varSigma$. (The token embedding function $\operatorname{emb}$ is just a formal description of `torch.nn.Embedding`.)
> >
> > > *Q4: L.158: You call: key map $\operatorname{qry}\_{l,k}$. It should be: $\operatorname{key}\_{l,k}$?*
> >
> > Thank you for pointing out this typo! We have fixed it in the revised paper.
> >
> > > *Q5: L. 335: why do you define E(0) := ALALA1 and not as E(0) := ALA0ALA1 ?*
> >
> > The two definitions are equivalent. Since all tape cells are initially filled with zeros, we do not have to explicitly use an A0 instruction to write an zero. We just used the shorter definition E(0) := ALALA1.
> >
> > References:
> > - [MS24] The expressive power of Transformers with chain of thought. *ICLR*, 2024.
> > - [PMB19] On the Turing completeness of modern neural network architectures. *ICLR*, 2019.
> > - [SS92] On the computational power of neural nets. *COLT*, 1992.
> > - [BSS89] On a theory of computation and complexity over the real numbers: NP-completeness, recursive functions and universal machines. *Bulletin of the AMS, 21*(1), 1989.
> > - [CKKLW95] On real Turing machines that toss coins. *STOC*, 1995.

---

> > > ### Comment · Reviewer_h1Bs · 2024-11-24
> > > **Precision complexity**
> > >
> > > Dear Authors,
> > >
> > > thank you for your thorough answers. Below you will find my further comments / questions.
> > >
> > > The main contribution of your work (Theorem 3.1) says that there exists a universal transformer. Importantly, as you confirmed in your rebuttal, you assume that arithmetic computations performed in the transformer are exact (i.e., infinite precision). However, this is not very surprising: enabling infinite precision + time, allows for huge computational power. For example, it is easy to see that even a simpler computational model of constant size —  a Random-access Machine (RAM) with a *finite number of registers* of infinite precision (or equivalently registers which can store integers) is able to solve the same problem as the transformer in Theorem 3.1. So, what is the main message of the theorem: a) from the point of view of computability and b) from ML?
> > >
> > > > The precision complexity of the Transformer is not about Turing completeness. It can be regarded as an analog of the space complexity of Turing machines --- even though a Turing machine is assumed to have infinite tapes, researchers are still interested in how much space is needed for a Turing machine to function.
> > >
> > > I do not agree with this. All you prove is that for a function computable in time t(n), the transformer form Theorem 3.1 requires precision complexity log(n+t(n)), which again one would expect. In fact, it is interesting and important to study, in addition to time, also the space complexity of problems. In a typical setting, one considers the space complexity S(n) of a TM, but allowing an arbitrary running time (up to exp(S(n))). So, in particular, PSPACE contains all problems solvable in polynomial space and time exp(n^{O(1)}), while EXTIME contains all problems solvable in time exp(n^{O(1)}). Given your setting, for both any problem in PSPACE and any problem in EXTIME your transformer computes a CoT of length exp(n^{O(1)}) and, at the same time, it requires precision poly(n). So, I don't see how precision complexity could affect space complexity in any way.
> > >
> > > Regards

---

> > > > ### Author Response · Authors · 2024-11-27
> > > > **Follow-up response (Part 1/2)**
> > > >
> > > > Thank you so much for your quick response and your interesting follow-up questions. We provide our response below.
> > > >
> > > > > *Q6: The main contribution of your work (Theorem 3.1) says that there exists a universal transformer. Importantly, as you confirmed in your rebuttal, you assume that arithmetic computations performed in the transformer are exact (i.e., infinite precision). However, this is not very surprising: enabling infinite precision + time, allows for huge computational power. For example, it is easy to see that even a simpler computational model of constant size — a Random-access Machine (RAM) with a finite number of registers of infinite precision (or equivalently registers which can store integers) is able to solve the same problem as the transformer in Theorem 3.1. So, what is the main message of the theorem: a) from the point of view of computability and b) from ML?*
> > > >
> > > > A) From the computability perspective, we would like to clarify that **even assuming infinite precision**, Transformers are still **a more restricted model of computation** than RAMs:
> > > >
> > > > - RAMs can override any registers, but **Transformers cannot override** any previous CoT steps or any internal values. This makes Transformers less powerful than RAMs. For example, a RAM with O(1) registers can compute the Parity function, but any Transformer with O(1) CoT steps **cannot compute the Parity function** [Hahn20].
> > > > - The operations that a Transformer can use are **much less flexible than RAMs**. This also makes it harder to explicitly construct a desired Transformer. For instance, as elaborated in Section 3.4, Transformers' attention operation is an average rather than a sum. Hence, it **cannot directly compute the prefix sum** (i.e., $\sum_{j=0}^iv_j$ for each $i$), which is very easy for a RAM. As a workaround, we have to compute **an indirect representation of the prefix sum** instead using Transformer operations.
> > > >
> > > > Due to the reasons above, we should not expect Transformers to be able to parse an arbitrary symbolic language. Therefore, it is **non-trivial to construct a Transformer and a corresponding "language"** such that (i) the Transformer is able to parse the "language" and that (ii) the "language" is powerful enough to efficiently achieve Turing completeness. That is why we need to specially propose our 2-PTMs (Section 3.1), our CoT step encoding (Section 3.2), and our input tokenizer (Section 3.3) as the "language."
> > > >
> > > > B) From the ML perspective, Theorem 3.1 shows that there exists **a single Transformer that can solve any (computable) task simply by prompting**. It is the **first theoretical underpinning** for the empirical success of "prompting."
> > > >
> > > > - Since the success of GPT, Transformer-based LLMs have revolutionized ML, shifting the classic **one-model-one-task** paradigm to the current **one-model-many-tasks** paradigm. The core of the one-model-many-tasks paradigm is **prompting**: to solve a a new task, people can simply describe the task in a prompt **without training a new model**.
> > > > - Despite the empirical success of LLMs, there is **no existing theory on "prompting" so far.** Existing ML expressiveness theories all focus on the classic **one-model-one-task** paradigm. That is, they show that given a task, there exists a model that solves this task. Unfortunately, these works **did not consider how to show the power of prompting** even though the expressiveness of Transformers has been well studied since 2019. We believe this is because this area currently lacks an appropriate theoretical framework to formalize what "prompting" is.
> > > > - To bridge this gap, we proposed our 2-PTM-based machinery to formalize prompting and, for the first time, showed that **there exists a single Transformer that can solve any (computable) task simply by prompting**. This is the first theoretical underpinning for the empirical success of prompting.
> > > >
> > > > Therefore, we believe that Theorem 3.1 is a non-trivial result from both the computability perspective and the ML perspective.

---

> > > > ### Author Response · Authors · 2024-11-27
> > > > **Follow-up response (Part 2/2)**
> > > >
> > > > > > *The precision complexity of the Transformer is not about Turing completeness. It can be regarded as an analog of the space complexity of Turing machines --- even though a Turing machine is assumed to have infinite tapes, researchers are still interested in how much space is needed for a Turing machine to function.*
> > > > >
> > > > > *Q7: I do not agree with this. All you prove is that for a function computable in time t(n), the transformer form Theorem 3.1 requires precision complexity log(n+t(n)), which again one would expect. In fact, it is interesting and important to study, in addition to time, also the space complexity of problems. In a typical setting, one considers the space complexity S(n) of a TM, but allowing an arbitrary running time (up to exp(S(n))). So, in particular, PSPACE contains all problems solvable in polynomial space and time $\exp(n^{O(1)})$, while EXTIME contains all problems solvable in time $\exp(n^{O(1)})$. Given your setting, for both any problem in PSPACE and any problem in EXTIME your transformer computes a CoT of length $\exp(n^{O(1)})$ and, at the same time, it requires precision poly(n). So, I don't see how precision complexity could affect space complexity in any way.*
> > > >
> > > > Firstly, sorry for the confusion. We did not mean that precision complexity affects space complexity or vice versa. We just wanted to **illustrate the motivation** of studying precision complexity by **making a conceptual analogy** between precision compleixty and space complexity.
> > > >
> > > > Apart from that, thank you for the interesting question on space complexity vs precision complexity. We agree that unlike PSPACE, our proof did not improve precision complexity by sacrificing other complexities. This is because our precision complexity **already matches the best known precision complexity** of the class of all Transformers [PBM21]. As a single Transformer is more restricted than the class of all Transformers, we do not expect to further improve this best known precision complexity.
> > > >
> > > > Regarding the relation between space complexity and precision complexity, we in fact conjecture that precision complexity of Transformers is **more closely related to time complexity than to space complexity.**  Our conjecture argument is as follows:
> > > >
> > > > 1. It is not hard to show that **CoT(poly(n)) is contained in P** using Theorem 3 of [MS24]. This implies that any problem **not in P requires super-polynomial** CoT steps. The intuition here is that unlike RAMs and TMs, **Transformers cannot override any previous CoT steps** (as we discussed in Q6).
> > > > 2. Note that O(log n) precision can represent at most $2^{O(\log n)}=n^{O(1)}=poly(n)$ positional encodings. Hence, if there are super-polynomial CoT steps, some steps must have identical positional encodings, making it difficult for the Transformer to distinguish those CoT steps. Although it is not a rigorous proof, this counting argument suggests that O(log n) precision is probably **not enough to handle super-polynomial CoT steps**.
> > > > 3. In particular, combining 1 and 2 implies that O(log n) precision is **not enough for any problem in PSPACE but not in P** (assuming that PSPACE is not equal to P). That is, small space complexity (i.e., PSPACE) may not imply small precision complexity (i.e., PREC(log n)). This suggests that precision complexity of Transformers is less related to space complexity than to time complexity.
> > > >
> > > > Therefore, we mainly focused on time complexity rather than space complexity in this work.
> > > >
> > > > That said, we still agree that studying the relation between precision complexity and space complexity and the optimal precision complexity will be very interesting future directions.
> > > >
> > > > Please feel free to let us know if you have further questions about our work.
> > > >
> > > > Thanks.
> > > >
> > > > -------------
> > > >
> > > > References:
> > > >
> > > > - [Hahn20] Theoretical limitations of self-attention in neural sequence models. Transactions of
> > > > the Association for Computational Linguistics, 8:156-171, 2020.
> > > > - [PBM21] Attention is Turing-complete. Journal of Machine Learning Research, 22(75):1–35, 2021.
> > > > - [MS24] The expressive power of Transformers with chain of thought. ICLR, 2024.

---

> > > > > ### Comment · Reviewer_h1Bs · 2024-12-02
> > > > > **Comment**
> > > > >
> > > > > Dear Authors,
> > > > >
> > > > > thank you for your answers and interesting comments. Indeed, given the fact that Transformers are able to analyze CoT but cannot replace any previous CoT step, one cannot expect any deeper relationship between precision and space complexity. But who knows? Anyway, the rebuttal addresses all my concerns. Thanks.

---

### Author Response · Authors · 2024-11-19
**General response**

We sincerely thank all reviewers for their valuable time and thoughtful reviews. We appreciate reviewers' recognition that a theoretical foundation of prompting is intersting (h1Bs) and important (cbcS), that the proof has a thorough construction (2gkz, k74c), and that the paper is well written (2gkz, cbcS). We also thank reviewers for their constructive comments, which will further improve the quality of the paper and lead to interesting future directions. We provide our response below and have revised our paper accordingly. We warmly invite the reviewers for further discussion.

---

### Meta-Review · Area_Chair_xxCW · 2024-12-18

**Metareview:**

This paper proves that LLM prompting is Turing complete, extending theoretical understanding to a common one-model-many-task setting of LLMs. I am not an expert in this area and lean on the reviewers for decision making. The reviewers appreciated the importance, novelty and technical correctness of the paper.

**Additional Comments On Reviewer Discussion:**

-

---

### Decision · Program_Chairs · 2025-01-22

Accept (Poster)